# Deficiency of the frontotemporal dementia gene *GRN* results in gangliosidosis

Sebastian Boland [1,2,11], Sharan Swarup[2,3,11], Yohannes A. Ambaw[1,2,4], Pedro C. Malia [1,2], Ruth C. Richards[1,2], Alexander W. Fischer[1,2], Shubham Singh[1,2], Geetika Aggarwal[5], Salvatore Spina[6], Alissa L. Nana [6], Lea T. Grinberg [6,7], William W. Seeley[6,7], Michal A. Surma [8], Christian Klose[8], Joao A. Paulo [2], Andrew D. Nguyen [5], J. Wade Harper [2,3,12] ✉, Tobias C. Walther [1,2,4,9,10,12] ✉ & Robert V. Farese Jr [1,2,4,10,12] ✉

Haploinsufficiency of *GRN* causes frontotemporal dementia (FTD). The *GRN* locus produces progranulin (PGRN), which is cleaved to lysosomal granulin polypeptides. The function of lysosomal granulins and why their absence causes neurodegeneration are unclear. Here we discover that PGRN-deficient human cells and murine brains, as well as human frontal lobes from *GRN*-mutation FTD patients have increased levels of gangliosides, glyco-sphingolipids that contain sialic acid. In these cells and tissues, levels of lysosomal enzymes that catabolize gangliosides were normal, but levels of bis(monoacylglycero)phosphates (BMP), lipids required for ganglioside cata-bolism, were reduced with PGRN deficiency. Our findings indicate that gran-ulins are required to maintain BMP levels to support ganglioside catabolism, and that PGRN deficiency in lysosomes leads to gangliosidosis. Lysosomal ganglioside accumulation may contribute to neuroinflammation and neuro-degeneration susceptibility observed in FTD due to PGRN deficiency and other neurodegenerative diseases.

About half of the human brain mass is composed of lipids[1]. Lysosomes are crucial in degrading and recycling cellular lipids, and accumulation of lipids and other macromolecules due to lysosomal dysfunction is linked to numerous neurodevelopmental and neurodegenerative dis-eases broadly classified as lysosomal storage disorders.

Granulins are polypeptides produced from progranulin (PGRN), a precursor protein that is cleaved in the lysosome. Deficiency of gran-ulins due to homozygous mutations in the *GRN* gene lead to neuronal ceroid lipofuscinosis[2], a severe neurodevelopmental disease, in humans and neuroinflammation in mice[3]. Haploinsufficiency of *GRN* almost invariably causes frontotemporal dementia (FTD)[4,5]. How granulins function in lysosomes and why their absence causes neuro-degeneration is unclear.

Inasmuch as granulin-deficiency is associated with lipofuscin accumulation, we tested the hypothesis that PGRN deficiency results in detrimental lysosomal lipid abnormalities. Consistent with this notion,

[1]Department of Molecular Metabolism, Harvard T. H. Chan School of Public Health, Boston, MA, USA. [2]Department of Cell Biology, Harvard Medical School, Boston, MA 02115, USA. [3]Aligning Science Across Parkinson's (ASAP) Collaborative Research Network, Chevy Chase, MD 20815, USA. [4]Center on Causes and Prevention of Cardiovascular Disease, Harvard T. H. Chan School of Public Health, Boston, MA 02115, USA. [5]Department of Internal Medicine, Division of Geriatric Medicine, and Department of Pharmacology and Physiology, Saint Louis University School of Medicine, St. Louis, MO 63104, USA. [6]Department of Neurology, Memory and Aging Center, University of California, San Francisco, San Francisco, CA 94158, USA. [7]Department of Pathology, University of California at San Francisco, San Francisco, CA, USA. [8]Lipotype GmbH, Dresden, Germany. [9]Howard Hughes Medical Institute, Boston, MA 02115, USA. [10]Broad Institute of Harvard and MIT, Cambridge, MA 02124, USA. [11]These authors contributed equally: Sebastian Boland, Sharan Swarup. [12]These authors jointly supervised this work: J. Wade Harper, Tobias C. Walther, Robert V. Farese. ✉e-mail: wade_harper@hms.harvard.edu; twalther@hsph.harvard.edu; robert@hsph.harvard.edu

a previous lipidomic study showed that PGRN deficiency in humans or mice alters levels of brain triglycerides (TAG), sterol esters (SE), and phosphatidylserine (PS)[6]. However, this study did not examine gangliosides, which are sialic-acid-containing glycosphingolipids that are highly abundant in the nervous system. Ganglioside catabolism occurs in the lysosome, and defects in the abundance or activity of different enzymes that catabolize gangliosides result in severe neurological diseases, such as Tay-Sachs or Sandhoff diseases[7,8] (Fig. 1a).

Here we discover that PGRN deficiency in cells, murine brains, or human frontal lobes of subjects with FTD due to *GRN* mutations, results in gangliosidosis. By studying this phenotype in human cells lacking PGRN, we uncover a mechanism for the gangliosidosis due to deficiency of lysosomal lipids that are required for ganglioside degradation. Our findings suggest a model for an adult neurodegenerative disease that may result from defective clearance of lysosomal lipids.

## Results

### Progranulin deficiency results in ganglioside accumulation in murine and human brains

We utilized lipidomics to examine glycosphingolipids in PGRN-deficient tissues and cells. We first analyzed brains isolated from 18-month-old *Grn*$^{R493X}$ mice, a murine model of PGRN deficiency[9]. These mice harbor the murine equivalent of the most prevalent human *GRN* mutation that causes FTD (R493X). They phenocopy *Grn* knockout mice, exhibiting CNS microgliosis, cytoplasmic TDP-43 accumulation, reduced synaptic density, lipofuscinosis, and excessive grooming behavior[9]. Lipidomics revealed increased levels of mono-sialylated GM1 species and a two- to four-fold increase in di-sialylated GD3 species in mouse whole cortex brains of *Grn*$^{R493X/R493X}$ mice (Fig. 1b, Supplementary Data 1). Levels of these gangliosides trended higher in *Grn*$^{+/R493X}$ heterozygous mice, but did not reach significance. GM2 levels also trended higher in *Grn*$^{R493X/R493X}$ mice, but were not significantly changed in *Grn*$^{+/R493X}$ mice. Di-sialylated GD1 species were increased in brains of *Grn*$^{+/R493X}$ brains and trended higher in *Grn*$^{R493X/R493X}$ brains (Fig. 1b, Supplementary Fig. 1a, and Supplementary Data 1). We also found modestly lower levels of long-chain bases (sphingosine and sphinganine) in *Grn*$^{R493X/R493X}$ brains than in control brains (Supplementary Fig. 1a). Because sphingosines are generated by degradation of more complex sphingolipids[10] (Fig. 1a), their reduced levels suggest that degradation of sphingolipids is impaired in PGRN-deficient brains. However, the levels of hexosylceramides (glucosylceramide and galactosylceramide) were similar in mouse brains for all genotypes (Supplementary Fig. 1a). Also, levels of the phospholipids phosphatidylethanolamine (PE), phosphatidylcholine (PC), PS, and of neutral lipids were comparable among genotypes (Supplementary Fig. 1a). Similar to the findings in the brain, deficiency of PGRN in the kidney also led to elevated levels of gangliosides (Supplementary Fig. 1b and Supplementary Data 1). However, ganglioside levels in rodent peripheral tissues are 2–10% of those in the brain[11], so the amount of gangliosides that accumulate in this tissue is considerably smaller.

To test whether PGRN deficiency's impact on the brain lipidome was also present in patients with GRN-related FTD, we analyzed the lipid composition of postmortem human frontal and occipital lobe brain tissue from control ($n = 3$), sporadic FTD (sporadic FTD-TDP, Type A, ($n = 6$), and *GRN* mutation-related FTD (GRN FTD-TDP, Type A, ($n = 12$) subjects (The number of available healthy control samples was unfortunately limited and therefore a limitation of these analyses). As reported for PGRN-deficient fibroblasts and murine brain[6], we found increased levels of SEs and TAGs in the frontal lobes of GRN FTD-TDP subjects; in contrast, TAGs were unchanged, and SEs were below the detection limit in the occipital lobes of the same subjects (Supplementary Fig. 1c and Supplementary Data 2). Additionally, we found reductions in PE and cardiolipins (Supplementary Fig. 1c) and increases

in sphingomyelin, particularly in the frontal lobes of the patients with GRN FTLD-TDP.

Human brains are abundant in a variety of gangliosides, including GM1, GD1a/b, GD3, and GT1b[12]. In a pattern that was similar to the changes in murine brain, we detected increased levels of mono-sialylated GM1 and di-sialylated GD3 and GD1 species in GRN FTD-TDP subjects (Fig. 1c and Supplementary Data 2). Some of these ganglioside species also trended higher in sporadic FTD-TDP subjects. The abundance of GT1, which can be catabolized at the plasma membrane[13], was lower in GRN FTD-TDP subjects and unchanged in sporadic FTD-TDP subjects (Supplementary Fig. 1c). In contrast to the findings in the frontal lobes, we detected no increase in the levels of gangliosides in the occipital lobes of either FTD group (Supplementary Fig. 1c and Supplementary Data 2).

### Progranulin-knockout HeLa cells accumulate GM2 gangliosides

To address the mechanism of ganglioside accumulation, we established HeLa TMEM192-3xHA cell lines (lysosomal tagged, ref. 14) with PGRN deficiency and such cells with PGRN expression restored (*GRN*$^{-/-}$ + PGRN-addback) via lentiviral transduction with untagged human PGRN cDNA (Fig. 2a and Supplementary Fig. 2a). In HeLa cells, the most abundant ganglioside class is GM2, and the gangliosides GM3, GD3, and GD1a[15] are present in lesser amounts. Levels of GM2 were ~two-fold higher in PGRN-knockout cells than control cells, and normal levels were restored by PGRN-addback. GD1 and GD3 levels were unchanged in PGRN-knockout cells, but GD3 levels were reduced in the PGRN-addback cells (Fig. 2b and Supplementary Data 3). TAG levels were increased in PGRN-knockout cells and restored by PGRN-addback, whereas PC, PE, and PS levels were unaffected upon PGRN depletion. For unclear reasons, diacylglycerol (DAG) levels (Fig. 2c) and several sphingolipid catabolic products were unchanged in PGRN-knockout cells but were increased in PGRN-addback cells (Supplementary Fig. 2b and Supplementary Data 3). The levels of unesterified cholesterol and cholesterol esters were similar in control, PGRN-knockout, and PGRN-addback cells, in contrast to the increased cholesterol and reduced cholesterol esters in NPC1- or NPC2-knockout cell lines that are deficient in cholesterol export from lysosomes (Supplementary Fig. 2c).

Next, we used immunostaining of PGRN-knockout cells with an antibody that detects GM2. In agreement with the increase in GM2 gangliosides detected by lipidomic analyses, we found that PGRN-knockout cells had more GM2 puncta than control cells, although less GM2 puncta than in positive control HEXA-deficient cells (Fig. 2d). These GM2 puncta partially co-localized with the lysosomal marker LAMP1, and their accumulation was reversed by PGRN-addback (Fig. 2d).

### Progranulin-knockout cells have normal amounts of glycosphingolipid catabolic enzymes but fewer lysosomal intralumenal vesicle-associated proteins

Gangliosides are catabolized by lysosomal enzymes, and deficiency of these enzymes in abundance or activity leads to lysosomal lipid accumulation. We, therefore, tested lysosomal function with a number of assays. Using quantitative whole-cell and lysosomal (isolated using Lyso-IP[14]) tandem mass tag (TMT) proteomics, we found no major differences in the abundances of lysosomal proteins or glycosphingolipid-metabolizing enzymes in PGRN-knockout and PGRN-addback cells (Fig. 3a–d, Supplementary Fig. 3a, and Supplementary Data 4 and 5). Moreover, the activity of the glycosphingolipid catabolism enzyme β-hexosaminidase subunit α (HEXA) was unchanged in PGRN-knockout and PGRN-addback genotypes when incubated with artificial substrates (Fig. 3e). The activity of glucosylceramidase β (GCase), another glycosphingolipid catabolism enzyme, was more variable in PGRN-knockout cell lysate than in control, and the average trended lower (-15%). Most of the lysosomal proteome was unaffected in mouse brains deficient for PGRN, but we found a modest

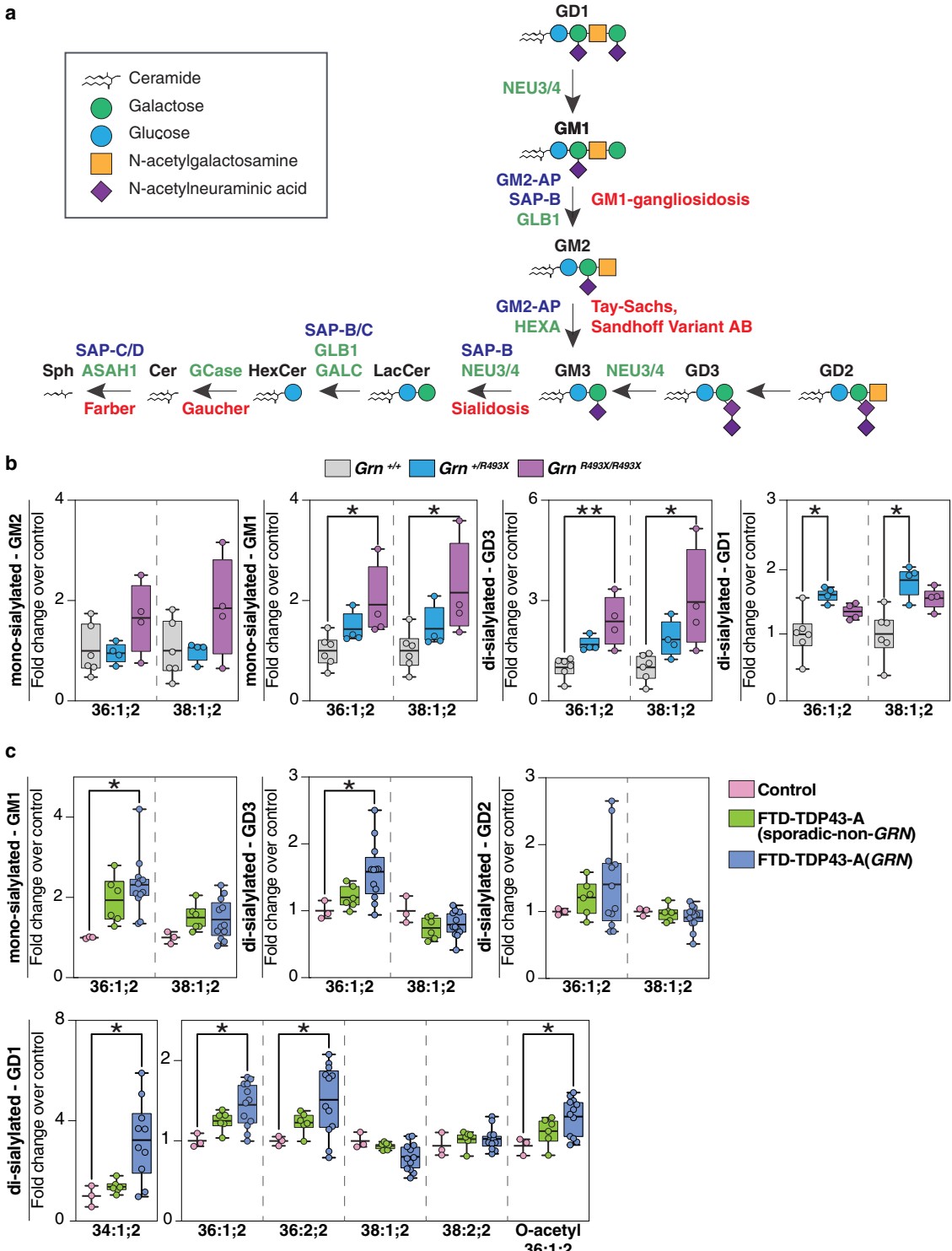

**Fig. 1 | Deficiency of progranulin leads to ganglioside accumulation in mouse and human brain tissues. a** Ganglioside degradation pathway in the lysosome. The names of glycosyl hydrolases (green), activator proteins (purple), and associated metabolic diseases (red) are indicated in the scheme. **b** Quantification of mono-sialyated and di-sialyated ganglioside species isolated from $Grn^{+/+}$ (gray) ($n = 6$), $Grn^{+/R493X}$ (blue) ($n = 4$), and $Grn^{R493X/R493X}$ (purple) ($n = 4$) mouse brains. **c** Quantification of mono-sialyated and di-sialyated ganglioside species isolated from the frontal lobes of control (pink), FTD-TDP43-A (sporadic-non-GRN) (green), and FTD-TDP43-A (GRN) (blue) human brains. Box plots display mean ± the minimum and maximum number in the data set of control ($n = 3$), FTD-TDP43-A

(sporadic-non-GRN) ($n = 6$) or FTD-TDP43-A (GRN) ($n = 12$). Box plots display mean ± the minimum and maximum number. One-way ANOVA, followed by multigroup comparison (Dunn's) test, was performed. *$p < 0.05$, **$p < 0.01$. G is for ganglioside; M/D/T are for monosialic, disialic, or trisialic; and the number refers to the order of discovery. GM2-AP GM2 ganglioside activator protein, SAP-B/C/D saposin-B/C/D, GLB1 galactosidase beta 1, HEXA beta-hexosaminidase subunit alpha, NEU3/4 neuraminidase 3/4, GALC galactosylceramidase, GCase glucosylceramidase beta, ASAH1 N-acylsphingosine amidohydrolase 1, SAP-C/D saposin-C/D.

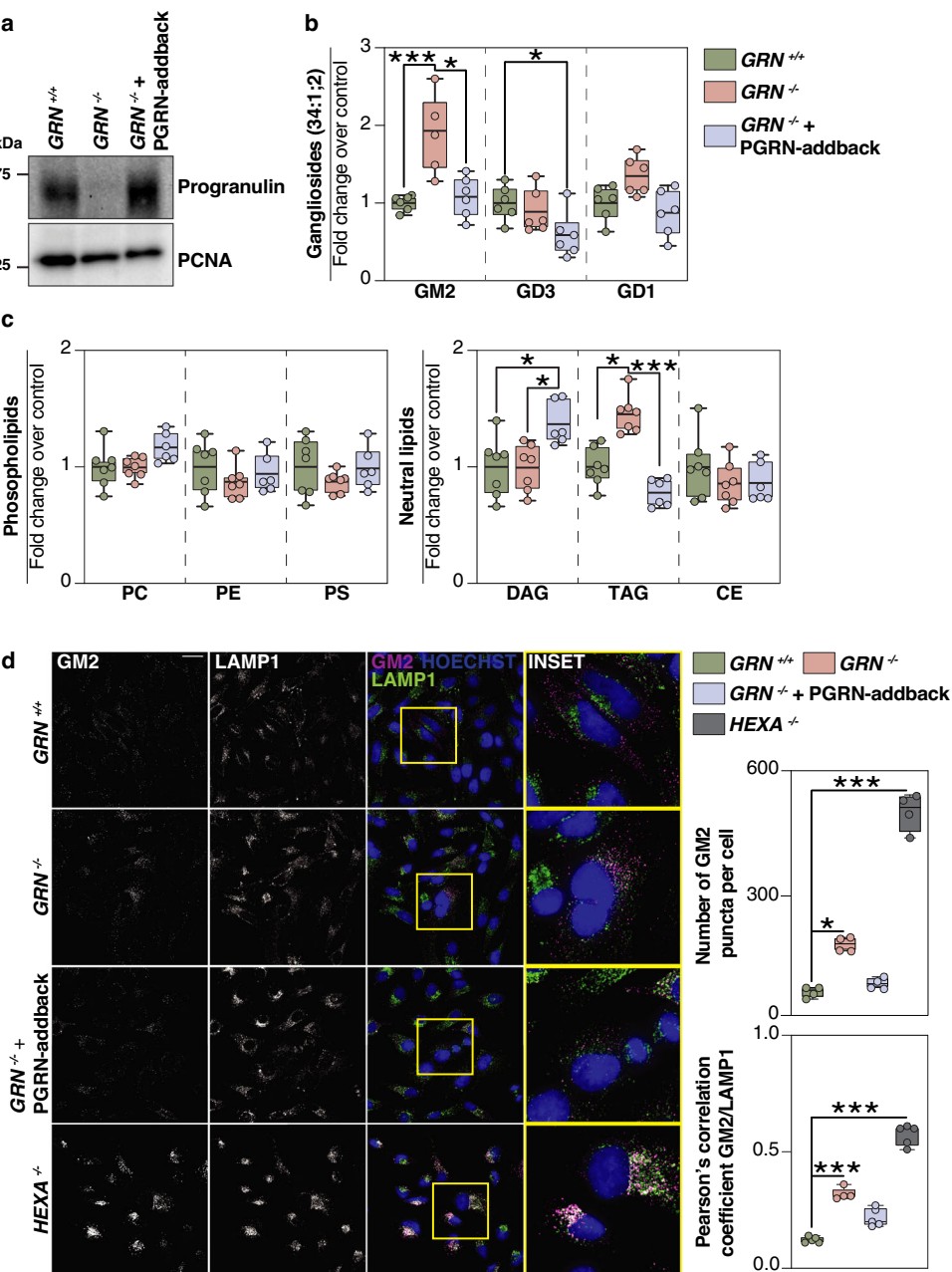

**Fig. 2 | Lipidomic and immunofluorescence analyses of HeLa cells reveals GM2 accumulation upon loss of progranulin that is restored by PGRN-addback.**
**a** Western blot of full-length progranulin protein levels in $GRN^{+/+}$, $GRN^{-/-}$, and $GRN^{-/-}$ + PGRN-addback HeLa cell lines. HeLa cells were gene-edited to contain TMEM192-3xHA. PCNA, proliferating cell nuclear antigen. **b** Quantification of gangliosides (GM2, GD3, GD1) and **c** quantification of phospholipids (PC, PE, PS) and neutral lipids (DAG, TAG, SE) isolated from $GRN^{+/+}$ (green) ($n = 7$), $GRN^{-/-}$ (orange) ($n = 7$), and $GRN^{-/-}$ + PGRN-addback (blue) ($n = 6$) HeLa cell lines.
**d** Representative confocal images of fixed HeLa cells stained with anti-GM2

antibody (magenta), anti-LAMP1 antibody (green) and Hoechst (blue). Scale bar, 50 µm. Bar graphs display number of GM2 puncta per cell and the Pearson's correlation coefficient between GM2/LAMP1. The numbers of cells used to calculate the GM2 puncta per cell were $n = 67$, $n = 69$, $n = 63$, $n = 42$ for $GRN^{+/+}$, $GRN^{-/-}$, $GRN^{-/-}$ + PGRN-addback, and $HEXA^{-/-}$, respectively. Box plots display mean ± the minimum and maximum number. One-way ANOVA, followed by multigroup comparison (Dunn's) test, was performed. *$p < 0.05$ or ***$p < 0.001$. PC phosphatidylcholine, PE phosphatidylthanolamine, PS phosphatidylserine, DAG diacylglycerol, TAG triacylglycerol, SE sterol esters.

upregulation of several enzymes that mediate glycosphingolipid degradation (Supplementary Fig. 3b and Supplementary Data 4). Furthermore, we found normal enzymatic activity of HEXA in murine and human brain protein lysates in vitro. Similar to cells, the activity of GCase measured in mouse brain lysates derived from $Grn^{R493X/+}$ and $Grn^{R493X/R493X}$ genotypes trended lower than in control brain lysate (Supplementary Fig. 3c). Notably, human brain lysate of the frontal lobe had less GCase activity when incubated with artificial substrates (Supplementary Fig. 3c). In contrast, brain lysate of the occipital lobe showed no differences in GCase activity (Supplementary Fig. 3c).

Upon screening for other lysosome-mediated processes, we found no differences for mTOR signaling, autophagic flux, or phosphorylation of microphthalmia/transcription factor E (MiT/TFE) proteins in PGRN-deficient cells or tissues (Supplementary Fig. 3d, e). These results suggest that PGRN depletion has minimal effects on lysosomal composition and function in HeLa cells under basal conditions.

Intralumenal vesicles (ILVs) are sites of lipid degradation within lysosomes. Electron microscopy used to analyze the ultrastructure of lysosomes revealed ILVs were present in control and PGRN-deficient

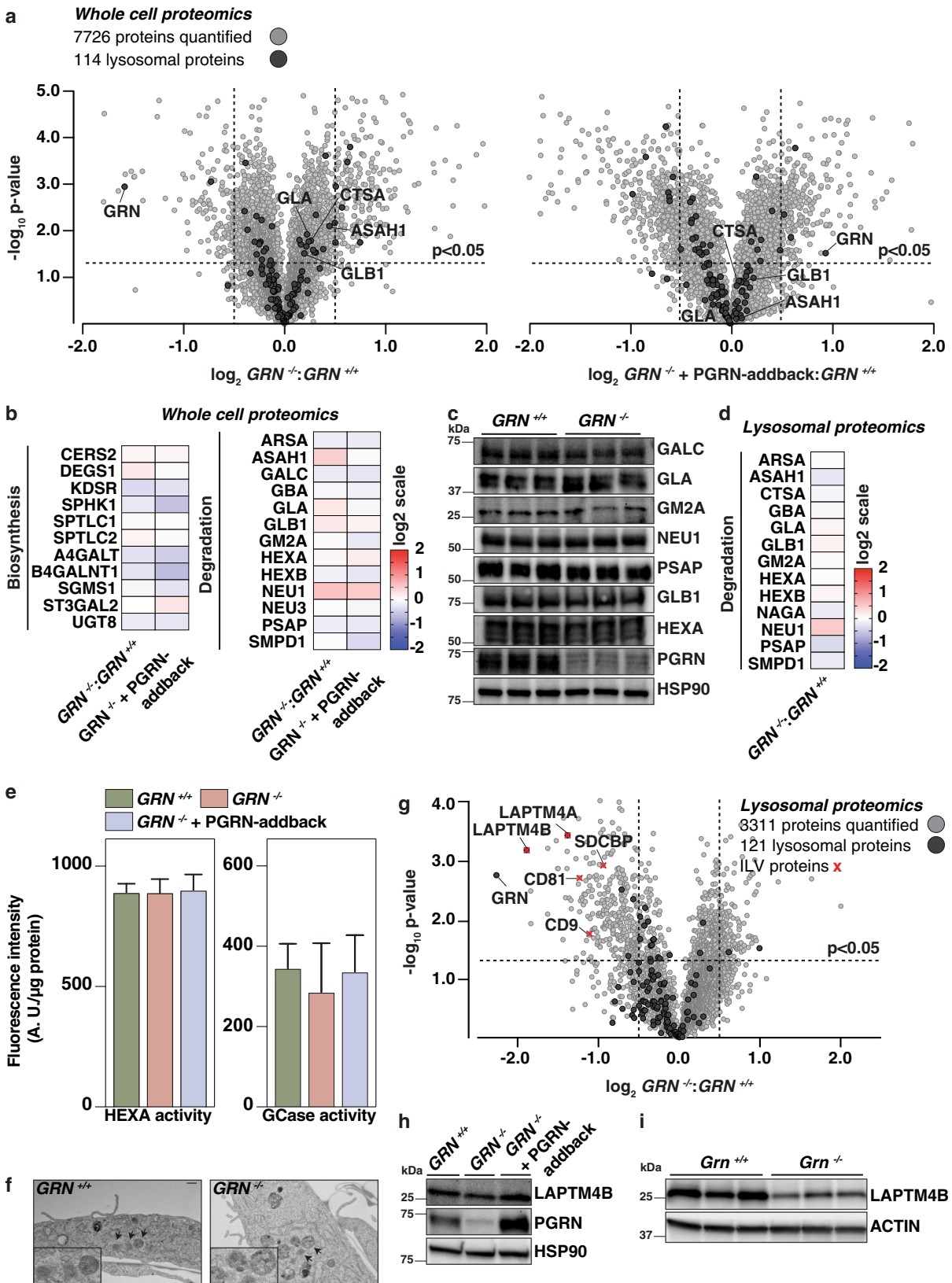

cells (Fig. 3f). However, lysosomes isolated from PGRN-deficient cells by Lyso-IP had fewer ILV-associated proteins, including LAPTM4A, LAPTM4B, SDCBP, CD81, and CD9 (Fig. 3g). Further, the modest reduction of LAPTM4B in PGRN-knockout cells was reversed by the expression of exogenous PGRN (Fig. 3h). Reduced levels of LAPTM4B were also found in brains of *Grn* knockout mice (Fig. 3i).

## Progranulin-knockout cells and tissues have reduced levels of BMP, and restoring BMP levels normalizes ganglioside levels

Because the levels of the enzymes that catabolize gangliosides were not changed in PGRN-deficient cells or tissues, and lysosomes appeared mostly intact and functional, we searched for another cause of gangliosidosis. A recent study found that

**Fig. 3 | TMT-quantitative proteomic and in vitro analyses show no major differences in abundances or activities of glycosphingolipid metabolic enzymes in cells with PGRN depletion. a** Volcano plot representation of whole-cell proteomes of $GRN^{-/-}$ (left) and $GRN^{-/-}$ + PGRN-addback (right) plotted against $GRN^{+/+}$ with log$_2$-fold-change (ratio of relative abundance, x-axis) and -log$_{10}$ p-value (y-axis). All proteins (gray) and lysosomal proteins (black) quantified are shown. A corrected p-value < 0.05 (Welch's test, two-sided) was used to calculate statistically significant differences between genotypes. **b** Heat map analysis of the relative abundance of a subset of proteins from whole-cell extracts that are involved in glycosphingolipid biosynthesis and degradation. **c** Western blotting analysis of proteins that are involved in glycosphingolid degradation (n = 3). **d** Heat-map analysis of the relative abundance of a subset of proteins that are involved in glycosphingolipid degradation from isolated lysosomal extracts. **e** HEXA and GCase activities were assessed in $GRN^{+/+}$ (green), $GRN^{-/-}$ (orange), and $GRN^{-/-}$ + P$GRN$-addback (blue) cells when incubated with artificial substrates (n = 4 with three technical replicates each, mean ± SD). One-way ANOVA, followed by multigroup comparison (Dunn's test), was performed. *p < 0.05, **p < 0.01, or ***p < 0.001. **f** Electron micrographs from $GRN^{+/+}$ and $GRN^{-/-}$ cells with lysosomes and ILVs indicated by arrows. Scale bar, 500 nm. **g** Volcano plot representation of lysosomal proteomes of $GRN^{-/-}$ plotted against $GRN^{+/+}$ with log$_2$-fold-change (ratio of relative abundance, x-axis) and −log$_{10}$ p-value (y-axis). All proteins (gray), lysosomal proteins (black), and ILV-associated proteins (red) quantified are shown. A corrected p-value < 0.05 (Welch's test, two-sided) was used to calculate statistically significant differences between genotypes. **h, i** Western blotting analysis of abundance of LAPTM4B in whole-cell extracts and mouse brains from different genotypes (n = 3).

bis(monoacylglycero)phosphate (BMP) levels are reduced in *Grn*-deficient mouse brains[16]. BMP is crucial in glycosphingolipid and ganglioside degradation in lysosomes[17], and its levels are altered in many lysosomal storage diseases[18]. BMP is found in ILVs where its negatively charged phosphate headgroup is thought to enable binding of lysosomal hydrolases[19,20]. We hypothesized that reduced BMP levels underlie the gangliosidosis we found in progranulin deficiency. We measured BMP levels in PGRN-knockout HeLa cells and found it ~50% reduced, whereas levels of the BMP isomer (and presumptive precursor) PG were unchanged. The reductions in BMP levels were restored in PGRN-addback cells (Fig. 4a and Supplementary Data 4). We also examined BMP accumulation in the HeLa cell model system with radioactive tracers. Metabolic labeling studies utilizing $^{14}$C-arachidonic acid confirmed the reduction of BMP levels in PGRN-knockout HeLa cells (Fig. 4b), suggesting alterations in the synthesis or degradation of BMP with polyunsaturated fatty acids in PGRN deficiency. Similarly, brains of PGRN-deficient mice showed a 50–60% reduction in BMP levels, and all detected BMP species were significantly reduced in *Grn* $^{R493X/R493X}$ brains (Fig. 4c, Supplementary Fig. 4a, and Supplementary Data 2).

We also measured BMP levels in brain tissues from patients with GRN FTD-TDP and compared them with sporadic FTD-TDP and control subjects. We found marked decreases in a BMP species containing two docosahexaenoic acid moieties (22:6/22:6) in the frontal and occipital lobes of all FTD subjects (Fig. 4d, Supplementary Fig. 4b, and Supplementary Data 1). BMP with two 22:6 polyunsaturated fatty acids is one of the most abundant BMP species in human brain[21]. BMP species with mono- or di-unsaturated fatty acid moieties trended lower in some samples but were not overall different among the study groups.

To further test if BMP deficiency is responsible for ganglioside accumulation in PGRN-knockout cells, we assayed whether adding exogenous BMP to HeLa cells lacking PGRN could normalize ganglioside levels. We incubated cells with di-oleoyl-PC or di-oleoyl-BMP and measured ganglioside levels by lipidomics. BMP but not PC was sufficient to normalize GM2 ganglioside levels in PGRN-knockout cells to levels similar to those in control cells (Fig. 4e). This finding was confirmed by feeding BMP to cells and measuring the number of GM2 puncta by immunofluoresence (Supplementary Fig. 4c, d). These experiments establish a causal link between BMP deficiency and ganglioside accumulation in PGRN-deficient cells.

## Discussion

Our experiments lead to a model for PGRN function in which lysosomal granulin peptides maintain lysosomal function and homeostasis, including the levels of BMP, that are crucial for ganglioside catabolism (Fig. 4f). In the setting of PGRN deficiency, reduced granulins in lysosomes lead to reduced BMP levels on intralumenal vesicles through an unclear mechanism. Low BMP levels, in turn, impair ganglioside catabolism and result in gangliosidosis. The resulting gangliosidosis may also compromise certain lysosomal functions, such as the response to membrane perturbation[16] and eventually lead to other hallmarks of the disease (e.g., protein aggregation of TMEM106B[22], TDP-43[23,24]). In agreement with an important role of lysosomal degradation in FTD, mutation of *CHMP2B*, a component of the ESCRT machinery of ILV formation, also leads to FTD[25]. Our model predicts that other lipids, such as neutral lipids or other sphingolipids, may also be poorly catabolized due to changes in lysosomal lipid degradation. Indeed, glucosylsphingosine, whose degradation is thought to require BMP, was reported to be increased in PGRN-deficient mice[16].

It is currently unclear how PGRN deficiency leads to low BMP levels. One possibility is that granulins may interact directly with BMP to modify its levels. A recent study reported that His-tagged PGRN bound to liposomes containing BMP[16], suggesting it may directly influence its abundance. We reproduced this finding for full-length progranulin in our binding assay (Supplementary Fig. 4e), but believe that further investigation for binding of native, untagged and functional granulin peptides is needed to understand this potential interaction and its implications. An alternative is that granulins indirectly affect BMP levels, for example by altering the ion environment of lysosomes or by changing synthesis or degradation rates of BMP through pathways that are currently not understood.

Our results are consistent with several reports linking PGRN deficiency to altered glycosphingolipid catabolism. Specifically, PGRN has been linked to the function of the HEXA enzyme[2], and GCase activity is compromised in PGRN-deficient mice and neurons[26,27]. Although we found that HEXA activity was not changed, GCase activity was decreased in the frontal lobes (but not the occipital lobes) of GRN FTD-TDP patients. Finally, and of particular relevance to our findings of PGRN deficiency causing gangliosidosis, a recent report found reduced BMP levels in brains of PGRN-deficient mice[16].

Despite the widespread expression of PGRN in tissues, PGRN deficiency results primarily in a disease of the central nervous system (CNS). Our findings suggest that this may be because gangliosides are found at much higher levels in the CNS[11] than in other tissues. In the brain, neuronal gangliosidosis may trigger the activation and recruitment of microglia to phagocytose and process the excess gangliosides. In particular, GM2 gangliosides may incite TNF-α expression and inflammation in monocyte-derived cells[28]. A hallmark of PGRN deficiency in murine brain is microgliosis and neuroinflammation[3], and microglial cells (and macrophages) are hyperactivated in the setting of PGRN deficiency[3]. The accumulated effects of long-term defects in lysosomal ganglioside metabolism and neuroinflammation may, therefore, contribute to GRN FTD-TDP.

With respect to therapeutic implications, increased ganglioside levels or reduced BMP levels may serve as biomarkers for PGRN-deficient FTD or other neurodegenerative disorders. It may also be of benefit to determine if drugs that lower ganglioside production[29] are beneficial in GRN-FTD-TDP. Finally, it will be of interest to analyze ganglioside levels in other chronic adult neurodegenerative diseases.

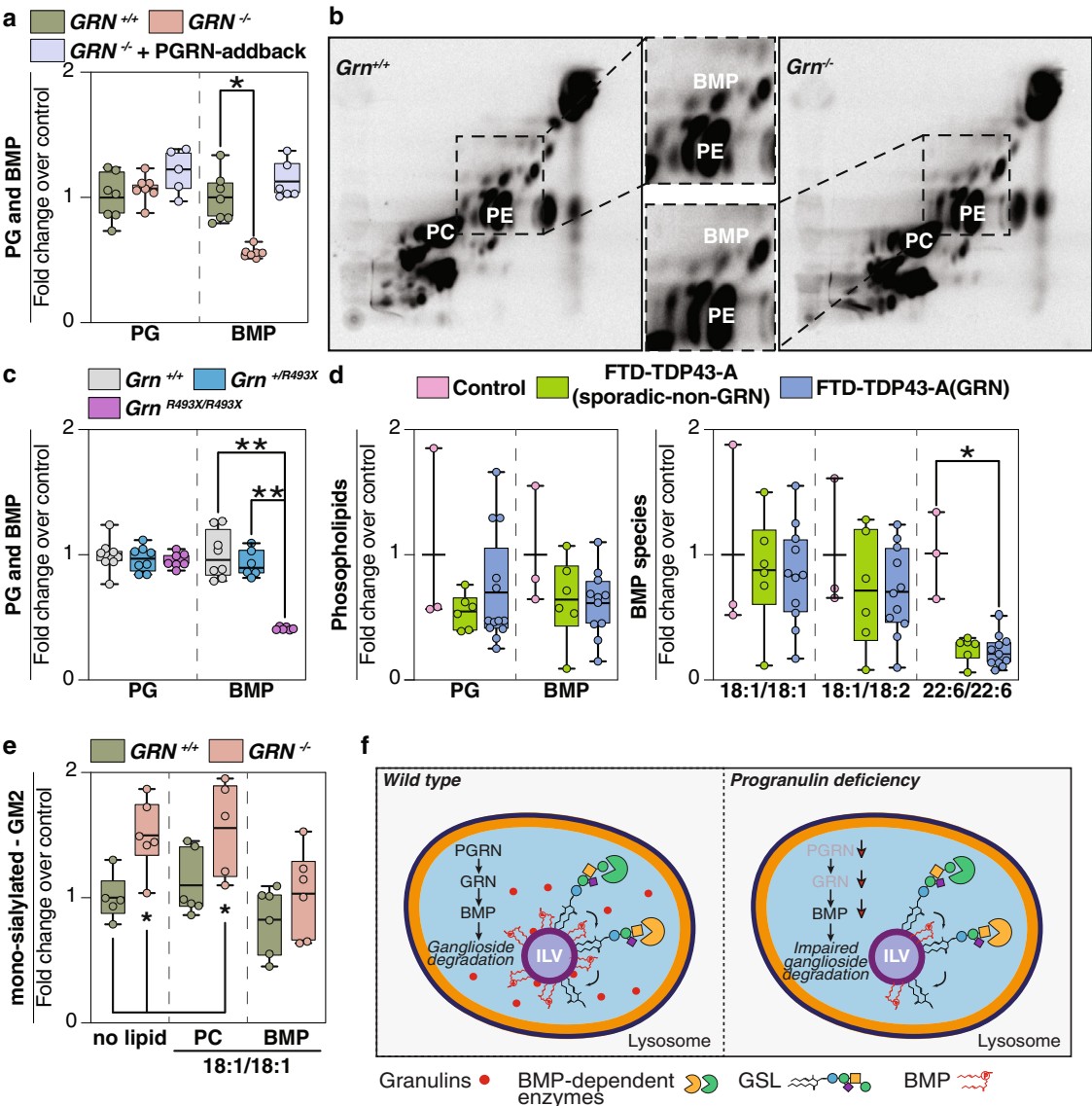

**Fig. 4 | BMP levels are reduced in progranulin-deficient cells or brain tissues.**
**a** Quantification of PG and BMP isolated from *GRN*⁺/⁺ (green), *GRN*⁻/⁻ (orange), and
*GRN*⁻/⁻ + PGRN-addback (blue) HeLa cell lines (*n* = 7) reveals a ~50% reduction in
BMP levels in PGRN-knockout cells. **b** Labeling of cellular lipids by feeding
¹⁴C-arachidonic acid–albumin complex to *GRN*⁺/⁺ and *GRN*⁻/⁻ HeLa cell lines
(60 min). Inset highlights reduced levels of BMP in *Grn*⁻/⁻ HeLa cells, compared to
control cells. **c** Quantification of PG isolated from *Grn*⁺/⁺ (gray) (*n* = 8), *Grn*⁺/R493X
(blue) (*n* = 8), and *Grn*R493X/R493X (purple) (*n* = 8) and quantification of BMP isolated
from *Grn*⁺/⁺ (gray) (*n* = 8), *Grn*⁺/R493X (blue) (*n* = 6), and *Grn*R493X/R493X (purple) (*n* = 6)
mouse brains reveals a decrease in BMP levels upon loss of PGRN in mouse brains
(-50%). **d** Quantification of PG and BMP isolated from the frontal lobes of control
(pink) (*n* = 3), FTD-TDP43-A(sporadic-non-GRN) (green) (*n* = 6), and FTD-TDP43-

A(GRN) (blue) (*n* = 11) human brains. BMP species with mono- or di-unsaturated
fatty acid moieties are not different, whereas BMP species containing two doc-
osahexanoic acid moieties (22:6/22:6) are reduced in the frontal and occipital lobes
of all FTD subjects. **e** Quantification of GM2 species isolated from *Grn*⁺/⁺ (green)
(*n* = 6) and *Grn*⁻/⁻ (orange) (*n* = 6) HeLa cells after feeding no lipids, di-oleoyl PC or
di-oleoyl BMP. **f** Model of the role of progranulin in the degradation of gangliosides.
PGRN/granulin deficiency leads to reduced BMP levels through unclear mechan-
isms. Reduced BMP levels contribute to impaired ganglioside degradation. Even-
tually, this leads to lysosomal dysfunction and downstream consequences,
including neuroinflammation and neurodegeneration. Box plots display mean ± the
minimum and maximum number. One-way ANOVA, followed by multigroup com-
parison (Dunn's test), was performed. *$p < 0.05$, **$p < 0.01$.

## Methods
### Chemicals and antibodies
The following reagents were purchased from commercial vendors:
acetonitrile, methanol, water (all HPLC/MS grade), chloroform (HPLC
grade, stabilized by 0.5–1% ethanol), ammonium formate, ammonium
acetate, formic acid, and acetic acid were purchased from Sigma-
Aldrich. N-Omega-CD3-octdecanoyl GM1 (Matreya LLC, Cat#2050),
SPLASH® LIPIDOMIX® Mass Spec Standard (Avanti Polar Lipids, Cat#
330707-1EA). TMTpro was purchased from Thermo Fisher
(Cat#A44520). Primary antibodies against the following targets were
used in the present study: GM2 monoclonal antibody (clone MK1-16)

(TCI America, Cat#A2576), human progranulin (R&D systems,
Cat#AF2420), anti-mouse progranulin polyclonal antibody that
recognizes an epitope at amino acids 198–214[30], PCNA (Santa Cruz
Biotechnology Cat#sc56), GALC (Proteintech Cat#11991-1-AP), GLA
(Proteintech Cat#66121-1-Ig), GM2A (Proteintech Cat#10864-2-AP),
NEU1 (Santa Cruz Biotechnology Cat#sc166824), PSAP (Proteintech
Cat#10801-1-AP), GLB1 (Proteintech Cat#15518-1-AP), HEXA (Pro-
teintech Cat#11317-1-AP), HSP90 (Proteintech Cat#60318-1-Ig), TFEB
(Cell Signaling Technology Cat#4240), TFE3 (Proteintech Cat#14480-
1-AP), SQSTM1 (Proteintech Cat#18420-1-AP), CALCOCO2 (Proteintech
Cat#12229-1-AP), MAP1LC3B (Cell Signaling Technology Cat#2775),

GABARAP (Proteintech Cat#18723-1-AP), MTOR (Cell Signaling Technology Cat#2983), MTOR (S2448) (Cell Signaling Technology Cat#2971), P70S6K (Cell Signaling Technology Cat#2708), P70S6K (T389) (Cell Signaling Technology Cat#9234), ULK1 (Cell Signaling Technology Cat#8054), ULK1 (S757) (Cell Signaling Technology Cat#14202), ASAH1 (Proteintech Cat#11274-1-AP), HEXB (Proteintech Cat#16229-1-AP) and BETA-ACTIN (Santa Cruz Biotechnology Cat#69879).

## Molecular cloning

The entry vector pDONR223 containing the full-length GRN (1–1179 base pairs) from the human orfeome collection was used to engineer a stop codon by site-directed mutagenesis (New England Biolabs) at the end of the open-reading frame sequence (ORF). Gateway technology (Thermo Fisher) was used to transfer the GRN ORF with LR cloning from the entry vector to the pHAGE lentiviral destination expression vector. sgRNA sequences for editing the *TMEM192*, *GRN*, *HEXA*, *NPC1* and *NPC2* loci were cloned into the pX459 V2.0 vector (Addgene Cat#62988) as described[31].

## Gene editing

CRSIPR/Cas9-mediated gene editing of HeLa cells (ATCC Cat# CCL-2) was performed as described (https://doi.org/10.17504/protocols.io.4r3l2oxqqv1y/v1)[31]. The following sgRNA sequences were used:

TMEM192 (5′-AGTAGAACGTGAGAGGCTCA-3′)
GRN (5′-ATCGACCATAACACAGCACG-3′)
HEXA (5′-CGGCCGAGCTGACATCGTAC-3′)
NPC1 (5′-TACCTGGACAGAAACTGTAG-3′)
NPC2 (5′-AGCTGCCAGGAAACGCATCG-3′)

To engineer the lyso-IP tag into HeLa cells by homology-directed repair, a gene block encoding a 3xHA epitope tag, a puromycin cassette, and homology arms on either side of the sgRNA cleavage site was synthesized by Integrated DNA Technologies to edit the *TMEM192* locus, similar to what was reported[14]. This sequence was cloned into the pSmart (Lucigen Cat#40041-2) donor vector using Gibson assembly (New England Biolabs). The donor vector along with the TMEM192 sgRNA sequence was transiently transfected into HeLa cells and puromycin selection was performed 5 days post-transfection for 7–8 days. The mixed pool of cells that were puromycin resistant were plated as single cells, and clonal lines of homozygous HeLa TMEM192-3xHA were isolated. The HeLa TMEM192-3xHA cell line was used for engineering all subsequent gene deletions.

## Lentivirus production

The lentiviral vector was packaged in HEK293T (ATCC Cat#CRL-3216) by co-transfection of psPAX2, pMD2.G (Addgene Cat#12260 Cat#12259) and pHAGE-GRN in a 4:2:1 ratio using polyethyleneimine. Virus-containing supernatant was harvested 2 days after transfection and filtered through a 0.45-micron syringe filter. Polybrene was added to a final concentration of 8 μg/ml to the viral supernatant. HeLa Tmem192-3xHA GRN KO cells were infected with 50 μL of viral supernatant, and stable cell lines were selected 48 h post-infection using hygromycin at a concentration of 100 μg/mL.

## Cell culture

HeLa TMEM192-3xHA and HEK293T cells were grown at 37 °C in Dulbecco' Modified Eagles Medium (DMEM) (Invitrogen Cat#11995-073), supplemented with 10% fetal bovine serum (FBS) (HyClone Cat#SH30910.03) and 1% penicillin-streptomycin (Thermo Fisher Cat#15140163).

## Immunofluorescence and imaging analysis

Cells were plated on to 24-well glass-bottom dish (Cellvis Cat#P24-1.5H-N). All immunofluorescence experiments were performed at room temperature. Cells at 70% confluence were washed twice in PBS

and fixed with 4% paraformaldehyde in PBS for 20 min. Cells were solubilized in 0.02% Saponin detergent in PBS for 15 min and then blocked with 2% BSA in 0.02% Saponin-PBS (blocking buffer) for 30 min. For the GM2-LAMP1 co-stain experiment, the cells were fixed and blocked, but not treated with detergent. Immunostaining for 2 h was performed with the following primary antibodies (1:100 dilution in blocking buffer): GM2 (Tokyo Chemical Industry Cat#2576), LAMP1 (Cell Signaling Technology Cat#9091), and HA-tag (Cell Signaling Technology Cat#3724). Cells were washed 3×5 min with PBS. Cells were incubated with the appropriate Alexa Fluor-conjugated secondary antibodies (Thermo Fisher Cat#A32731 Cat#A21203) (1:400 dilution in blocking buffer) for 1 h. Cells were washed 3 × 5 min with PBS. Cells were stained with Hoechst (1 μg/mL in blocking buffer) for 5 min. Cells were washed 3 × 5 min with PBS. Cells were imaged using a Yokogawa CSU-X1 spinning-disk confocal on a Nikon Ti-E inverted microscope at the Nikon Imaging Center in Harvard Medical School. The microscope is equipped with a Nikon Plan Apo 40x/1.30 NA objective lens and 445 nm (75 mW), 488 nm (100 mW), 561 nm (100 mW), and 642 nm (100 mW) laser lines controlled by AOTF. All images were collected with a Hamamatsu ORCA-ER cooled CCD camera (6.45 μm² photodiode) with MetaMorph image acquisition software. Z series are displayed as maximum z-projections and brightness and contrast were adjusted for each image equally and then converted to RGB. Image analysis was performed using Fiji[32].

## Western blotting analysis

General protocols for western blotting performed here can be found at: https://doi.org/10.17504/protocols.io.kxygxzr94v8j/v1. Cell pellets or mouse tissues were resuspended in ice-cold 8 M urea buffer (8 M urea, 50 mM Tris pH 7.4, 50 mM NaCl) supplemented with protease and phosphatase inhibitors (Roche). The resuspended samples were sonicated, and the lysates were clarified at 17000 x g for 10 min at 4 °C. A Bradford assay was performed, and equal amounts of lysate were boiled in LDS-Laemmli buffer supplemented with 50 mM DTT for 10 min at 95 °C. Lysates were run on 4–20% Tris glycine gels (BioRad) and transferred on to PVDF membranes (Millipore), which were blocked for 1 h at room temperature in 2% BSA-0.1% TBS-tween (blocking buffer). Immunoblotting with primary antibodies was performed overnight at 4 °C (1:1000 dilution in blocking buffer). Immunoblots were incubated with the appropriate secondary antibodies (1:5000 dilution in blocking buffer) for 1 h at room temperature. Images of blots were acquired using Enhanced-Chemi luminescence on a BioRad ChemiDoc imager.

## Pull-down assay

Cells pellets were harvested, washed twice in ice-cold PBS, and resuspended either in ice-cold neutral lysis buffer (50 mM Tris pH 7.4, 150 mM NaCl, 0.5% NP-40, pH 7.4) or acidic lysis buffer (50 mM NaOAc pH 5.3, 150 mM NaCl, 0.5% NP-40, pH 5.2) supplemented with protease and phosphatase inhibitors (Roche). All subsequent steps were performed at 4 °C. Resuspended cells were incubated on a rotator for 30 min, centrifuged at 17,000 × g for 10 min, and the supernatant was collected. In all, 1 mg of supernatant from each sample was incubated with 20 μL of BMP-beads, LPA-beads, or control beads (Echelon Biosciences Cat#P-BLBP-2 Cat#L-6101 Cat#P-B000) for 3 h on a rotator. The beads were pelleted by centrifugation at 500 × g for 1 min, the supernatant aspirated, and washed twice with either neutral or acidic lysis buffer. The beads were boiled in LDS-Laemmli buffer supplemented with 50 mM DTT for 10 min at 95 °C, and western blotting was performed as described earlier.

## Lipid-feeding experiment

Cells were plated on to six-well dish to 70% confluence. Next, medium was changed to DMEM supplemented with 10 μM 18:1/18:1-PC or 18:1/18:1-BMP. Lipid-supplemented medium was prepared by drying

the lipids in a glass vial under $N_2$ stream, and dried lipids were resuspended in complete DMEM medium using water bath sanctum (~30 min) to get the final lipid concentration to 10 µM lipids in the medium. Cells were maintained in lipid-supplemented medium for 24 h, and medium was changed every 6 h to ensure lipid supplementation. Cells were washed with cold PBS and processed for lipid extraction or fixed with 4% (w/v) of formaldehyde in PBS for anti-GM2 ganglioside immunofluorescence assay.

## Electron microscopy imaging

Electron microscopy imaging was performed in the Harvard Medical School Electron Microscopy Facility. Fixative solution containing 2.5% glutaraldehyde, 1.25% paraformaldehyde, 0.03% picric acid in 0.1 M sodium cacodylate buffer, pH 7.4, was added in a 1:1 ratio to cells grown to 70% confluency for 1 h at room temperature. The cells were then postfixed for 30 min in 1% osmium tetroxide ($OsO_4$)/1.5% potassium ferrocyanide ($KFeCN_6$), washed in water 3x, and incubated in 1% aqueous uranyl acetate for 30 min, followed by two washes in water and subsequent dehydration in grades of alcohol (5 min each; 50, 70, 95%, 2 × 100%). Cells were removed from the dish in propyleneoxide, pelleted at 500×$g$ for 3 min, and infiltrated for 2 h to overnight in a 1:1 mixture of propyleneoxide and TAAB Epon (Marivac Canada Inc., St. Laurent, Canada). The samples subsequently embedded in TAAB Epon and polymerized at 60 °C for 48 h. Ultrathin sections (~60 nm) were cut on a Reichert Ultracut-S microtome, picked up on to copper grids stained with lead citrate and examined in a JEOL 1200EX Transmission electron microscope or a TecnaiG² Spirit BioTWIN and images were recorded with an AMT 2k CCD camera.

## Lysosome purification

Lysosome immunoprecipitation was carried out as described[14] with a few modifications. All steps of the process were carried out at 4 °C with cold solutions. Briefly, HeLa TMEM192-3xHA endogenously tagged cells that were grown to 80% confluency in 150-mm plates were washed twice with PBS, scraped into tubes, and then pelleted at 500×$g$ for 5 min. The cells were re-suspended in 2 mL of lysoIP buffer (50 mM KCl, 100 mM $KH_2PO_4$, 100 mM $K_2HPO_4$, pH 7.2) supplemented with protease and phosphatase inhibitors (Roche). The cells were transferred to a glass homogenizer and dounced using 25 strokes. The lysed cells were centrifuged at 1000×$g$ for 10 min, and the post-nuclear supernatant (PNS) was collected. The concentration of the PNS was determined by Bradford assay. Lysosomes from normalized amounts of PNS were immunoprecipitated by incubation with 50 µL of magnetic HA-beads (Thermo Scientific Cat#88837) for 1 h on a rotator. The beads were sequentially washed once with lysoIP buffer containing 300 mM NaCl and once with lysoIP buffer. The lysosomes were solubilized and eluted off the beads by incubating the beads with 150 µL of lysoIP buffer with 0.5% NP-40 in a thermomixer (1000 rpm) for 30 min. The eluates were snap frozen in liquid nitrogen and stored in −80 °C.

## Mice

Animal procedures were approved by the Institutional Animal Care and Use Committee of the Harvard Medical Area Standing Committee on Animals and followed NIH guidelines. All mouse experiments were performed under the oversight and ethical guidelines from the Harvard Center for Comparative Medicine. Mice were housed in a pathogen-free barrier facility with a 12 h light/12 h dark cycle and allowed food and water ad libitum. $Grn^{-/-}$ mice[3] and $Grn^{R493X}$ mice[9] were on the C57BL/6 J background (backcrossed more than eight generations). Mice used in this study were aged 18–20 months and of both sexes.

## Human brain tissue studies

Postmortem brain samples were provided by the University of California, San Francisco (UCSF) Neurodegenerative Disease Brain Bank.

Brains were donated with the consent of the participants or their surrogates in accordance with the Declaration of Helsinki, and the research was approved by the University of California, San Francisco Committee on Human Research. Tissue blocks were dissected from the middle frontal gyrus and lateral occipital cortex of three controls, as well as six patients with sporadic FTLD-TDP and 13 with GRN-FTLD-TDP. All patients with FTD-$GRN$ carried a pathogenic variant in $GRN$ and had FTLD-TDP, Type A, identified at autopsy (Table see below). Clinical and neuropathological diagnoses were made using standard diagnostic criteria[33–37]. Characteristics of each group were as follows:

|  | Median age (years) | Interquartile ranges (years) |
| --- | --- | --- |
| Control ($n = 3$) | 86 | 11 |
| Sporadic FTD-TDP ($n = 6$) | 71 | 5 |
| GRN FTD-TDP ($n = 13$) | 68 | 2.5 |

Patients in the GRN-FTLD and sporadic FTLD groups showed a range of FTD syndromes (bvFTD, nfvPPA, etc.) and all showed FTLD-TDP Type A pathology (with one showing advanced comorbid AD).

## Lipidomics of the organic phase derived from cells and mouse tissues

HeLa cells were grown in a 10-cm culture dish until they reached ~80% confluence. Tissue samples were obtained as described[9], and mouse whole cortex brains were collected and immediately snap frozen in liquid nitrogen. Cell and tissue homogenates were obtained by snap-freezing (in liquid nitrogen)/thawing (using an ultrasound water bath for 3 min) repeatedly, and finally extracted according to Folch's method[38]. Internal standard SPLAH mix and deuterated ganglioside standard spiked in prior to extraction were used for normalization. The organic phase of each cell-culture sample was normalized by total soluble protein amounts and measured by BCA assay (Thermo Scientific, 23225, Waltham, MA), whereas tissue samples were normalized according to dry weight measurements. Samples were routinely subjected to two rounds of extraction.

The HPLC-mass spectroscopy (MS) method was adopted from[39]. Briefly, HPLC analysis of the organic phases was performed employing a C30 reverse-phase column (Thermo Fisher Scientific, Acclaim C30, 2.1 × 250 mm, 3 µm, operated at 55 °C; Bremen, Germany) connected to a Dionex UltiMate 3000 HPLC system and a QExactive orbitrap mass spectrometer (Thermo Fisher Scientific, Bremen, Germany) equipped with a heated electrospray ionization (HESI) probe. Dried lipid samples were dissolved in appropriate volumes of 2:1 MeOH:$CHCl_3$ (v/v), and 5 µL of each sample was injected, with separate injections for positive and negative ionization modes. Mobile phase A consisted of 60:40 can:$H_2O$, including 10 mM ammonium formate and 0.1% formic acid, and mobile phase B consisted of 90:10 2-propanol:ACN, also including 10 mM ammonium formate and 0.1% formic acid. The elution was performed with a gradient of 90 min; for 0–7 min, elution started with 40% B and increased to 55%; from 7 to 8 min, increased to 65% B; from 8 to 12 min, elution was maintained with 65% B; from 12 to 30 min, increased to 70% B; from 30 to 31 min, increased to 88% B; from 31 to 51 min, increased to 95% B; from 51 to 53 min, increased to 100% B; during 53 to 73 min, 100% B was maintained; from 73 to 73.1 min, solvent B was decreased to 40% and maintained for another 16.9 min for column re-equilibration. The flow-rate was set to 0.2 mL/min. The column oven temperature was set to 55 °C, and the temperature of the autosampler tray was set to 4 °C. The spray voltage was set to 4.2 kV, and the heated capillary and the HESI were held at 320 °C and 300 °C, respectively. The S-lens RF level was set to 50, and the sheath and auxiliary gas were set to 35 and 3 units, respectively. These conditions were held constant for both positive and negative ionization mode

acquisitions. External mass calibration was performed using the standard calibration mixture every 7 days.

MS spectra of lipids were acquired in full-scan/data-dependent MS2 mode. For the full-scan acquisition, the resolution was set to 70,000, the AGC target was 1e6, the maximum injection time was 50 msec, and the scan range was $m/z = 133.4–2000$. For data-dependent MS2, the top 10 ions in each full scan were isolated with a 1.0 Da window, fragmented at a stepped normalized collision energy of 15, 25, and 35 units, and analyzed at a resolution of 17,500 with an AGC target of 2e5 and a maximum injection time of 100 msec. The underfill ratio was set to 0. The selection of the top 10 ions was subject to isotopic exclusion with a dynamic exclusion window of 5.0 sec. Processing of raw data was performed using LipidSearch software (Thermo Fisher Scientific/Mitsui Knowledge Industries)[40,41].

## Lipidomics of the aqueous phase derived from cells and tissues

The aqueous phase was desalted by applying to a C18 cartridge (Waters) equilibrated with 2:43:55 (chloroform:methanol:water) and eluted with 1:1 CHCl3:MeOH. The eluates were dried down again and resuspended in chloroform:methanol:water (600:425:75, v/v/v).

The HILIC-MS method was adopted from[42]. HPLC analysis was performed employing a Phenomenex (Thermo Fisher Scientific, CAT#, 2.0 × 150 mm, operated at 60 °C; Bremen, Germany). Dried lipid samples were dissolved in appropriate volumes of 2:1 MeOH: CHCl₃ (v/v) and 5 μL of each sample was injected and acquired in negative ionization mode. Mobile phase A consisted of acetonitrile with 0.2% formic acid and mobile phase B consisted of 10 mM aqueous ammonium acetate, pH 6.1, adjusted with formic acid. Column equilibration was performed using 12.3% B for 5 min prior to each run. Chromatographic condition: mobile-phase gradient as follows: 0 min: 87.7% A + 12.3% B; and 15 min: 77.9% A + 22.1% B. The re-equilibration time between runs was 5 mins. The flow rate for the separation was set to 0.6 mL/min. The column oven temperature was set to 40 °C, and the temperature of the autosampler tray was set to 4 °C. The spray voltage was set to −4.5 kV, and the heated capillary and the HESI were held at 300 °C and 250 °C, respectively. The S-lens RF level was set to 50, and the sheath and auxiliary gas were set to 40 and 5 units, respectively. These conditions were held constant during the acquisitions.

External mass calibration was performed using the standard calibration mixture every 7 days. MS spectra of lipids were acquired in full-scan/data-dependent MS2 mode. For the full-scan acquisition, the resolution was set to 70,000, the AGC target was 1e6, the maximum injection time was 50 msec, and the scan range was $m/z = 700–2500$ in the negative ion mode. For data-dependent MS2, the top 10 ions in each full scan were isolated with a 1.0 Da window, fragmented at a stepped normalized collision energy of 25, 35, and 50 units, and analyzed at a resolution of 17,500 with an AGC target of 2e5 and a maximum injection time of 100 msec. The underfill ratio was set to 0. The selection of the top 10 ions was subject to isotopic exclusion with a dynamic exclusion window of 5.0 sec. Processing of raw data was performed in Xcalibur™ software (Thermo Fisher Scientific).

## Lipid extraction for mass spectrometry lipidomics of human brain samples

MS-based lipid analysis was performed by Lipotype GmbH (Dresden, Germany) as described[43]. If not indicated otherwise, 500 μg of tissue were used per extraction. Lipids were extracted using a two-step chloroform/methanol procedure[44]. Samples were spiked with internal lipid standard mixture containing: cardiolipin 16:1/15:0/15:0/15:0 (CL, 50 pmol per extraction), ceramide 18:1;2/17:0 (Cer, 30 pmol), diacylglycerol 17:0/17:0 (DAG, 100 pmol), hexosylceramide 18:1;2/12:0 (HexCer, 30 pmol), lyso-phosphatidate

17:0 (LPA, 30 pmol), lyso-phosphatidylcholine 12:0 (LPC, 50 pmol), lyso-phosphatidylethanolamine 17:1 (LPE, 30 pmol), lyso-phosphatidylglycerol 17:1 (LPG, 30 pmol), lyso-phosphatidylinositol 17:1 (LPI, 20 pmol), lyso-phosphatidylserine 17:1 (LPS, 30 pmol), phosphatidate 17:0/17:0 (PA, 50 pmol), phosphatidylcholine 17:0/17:0 (PC, 150 pmol), phosphatidylethanolamine 17:0/17:0 (PE, 75 pmol), phosphatidylglycerol 17:0/17:0 (PG, 50 pmol), phosphatidylinositol 16:0/16:0 (PI, 50 pmol), phosphatidylserine 17:0/17:0 (PS, 100 pmol), cholesterol ester 20:0 (CE, 100 pmol), sphingomyelin 18:1;2/12:0;0 (SM, 50 pmol), triacylglycerol 17:0/17:0/17:0 (TAG, 75 pmol), GM1-D3 18:1;2/18:0;0 (200 pmol), and cholesterol D6 (Chol, 300 pmol). After extraction, the organic phase was transferred to an infusion plate and dried in a speed vacuum concentrator. First, dry extract was re-suspended in 7.5 mM ammonium acetate in chloroform/methanol/propanol (1:2:4, V:V:V), and second, dry extract in 33% ethanol solution of methylamine in chloroform/methanol (0.003:5:1; V:V:V). All liquid handling steps were performed using Hamilton Robotics STARlet robotic platform with the Anti Droplet Control feature for organic solvents pipetting.

## Mass spectroscopy data acquisition of human brain samples

Samples were analyzed by direct infusion on a QExactive mass spectrometer (Thermo Scientific), equipped with a TriVersa NanoMate ion source (Advion Biosciences). Samples were analyzed in both positive and negative ion modes with a resolution of Rm/z = 200 = 280,000 for MS and Rm/z = 200 = 17,500 for MS/MS experiments, in a single acquisition. MS/MS was triggered by an inclusion list encompassing corresponding MS mass ranges scanned in 1 Da increments[45]. Both MS and MSMS data were combined to monitor CE, DAG and TAG ions as ammonium adducts; PC, PC, and O-, as acetate adducts; and CL, PA, PE, PE O-, PG, PI, and PS as deprotonated anions. MS only was used to monitor LPA, LPE, LPE O-, LPI, LPS, and GM2 as deprotonated anions; Cer, HexCer, SM, LPC, and LPC O-as acetate adducts; and cholesterol as an ammonium adduct of an acetylated derivative[46].

## Lipidomic analysis of gangliosides of human brain samples

Ganglioside classes GM1, GD1, GD2, GD3, GT1, GT2, GT3, and GQ1 were extracted and analyzed as follows. Gangliosides in the remaining water phase of the two-step chloroform:methanol procedure were subjected to purification using solid-phase extraction (Thermo Scientific SOLA SPE plates, 10 mg/2 mL)[47]. The water phase was loaded on columns pre-washed with chloroform:methanol (2:1, V:V), methanol and methanol:water (1:1, V:V); with the input flow through re-applied three times. Then, columns were washed with water, and the elution was carried out two times with methanol and one time with chloroform:methanol (1:1, V:V). Washing and elution steps were carried using a vacuum manifold. Pooled eluates were dried in a speed vacuum concentrator and re-suspended in 33% ethanol solution of methylamine in chloroform:methanol (0.003:5:1; V:V:V). Ganglioside extracts were analyzed by direct infusion on a QExactive MS (Thermo Scientific) equipped with a TriVersa NanoMate ion source (Advion Biosciences). Samples were analyzed in negative ion modes with a resolution of Rm/z = 200 = 140,000; AGC target of 1e6; maximum injection time of 500 ms and 3 microscans.

## Data analysis and post-processing of human brain samples

Data were analyzed with in-house developed lipid identification software, based on LipidXplorer[48,49]. Data post-processing and normalization were performed using an in-house developed data management system. Only lipid identifications with a signal-to-noise ratio >5, and a signal intensity five-fold higher than in corresponding blank samples were considered for further data analysis. Lipids were normalized to lipid class–specific internal standards. In case of

ganglioside classes for which no suitable lipid class–specific internal standards are available, spectral intensities were normalized to the internal standard GM1-D3 18:1;2/18:0;0, and the normalized intensities further normalized to total lipid content (in pmol) of the sample.

Only results with >1.5-fold-change and a *p*-value < 0.05 were selected from the web browser-based data visualization tool (LipotypeZoom) for levels of lipid species as the percentage of the total lipid amount for each sample.

## Enzyme activity assays

Protein estimation of the lysate was done by BCA assay. The activities of β-hexosaminidase and glucosylceramidase β were measured by a fluorometric method using 4-methylumbelliferyl N-acetyl-β-D-glucosaminide (Sigma, Cat#69585) and 4-methylumbelliferyl β-D-glucopyranoside (Sigma, Cat#M3633) as substrates, respectively. The reaction mixture consisted of either of the substrates (5 mM), enzyme fraction, and 100 mM citrate buffer, pH 5.0, in a final volume of 200 μL were incubated at 45 °C. After 1 h, the reaction was stopped with 2.5 mL of 0.5 M $Na_2CO_3$ buffer, pH 10.4. The 4-methylumbelliferone released was measured in a Tecan Spark multimode microplate reader with excitation and emission set at 350 and 440 nm, respectively.

## Proteomics - sample digestion

Detailed methods for proteomic analysis are provided at https://doi.org/10.17504/protocols.io.ewov14b7kvr2/v3. In all, 50 μg of protein extracts from cell pellets or mouse tissue were subjected to disulfide bond reduction with 5 mM TCEP (room temperature, 10 min) and alkylation with 25 mM chloroacetamide (room temperature, 20 min), followed by methanol–chloroform precipitation. For lysosomal samples, disulfide bond reduction and alkylation were performed using 20 μg of extracts, followed by trichloroacetic acid precipitation. Samples were resuspended in 50 μL of 200 mM EPPS, pH 8.5, and digested at 37 °C for 2 h with LysC protease at a 200:1 protein-to-protease ratio. Trypsin was then added at a 100:1 protein-to-protease ratio, and the reaction was incubated for 6 h at 37 °C. Tandem mass tag labeling of each sample was performed by adding 10 μL each of the 20 ng/μL stock of TMTpro reagent along with acetonitrile to achieve a final acetonitrile concentration of approximately 30% (v/v). After incubation at room temperature for 1 h, the labeling efficiency of a small aliquot was tested, and the reaction was then quenched with hydroxylamine to a final concentration of 0.5% (v/v) for 15 min. The TMTpro-labeled samples were pooled together at a 1:1 ratio. The sample was vacuum centrifuged to near dryness, resuspended in 5% formic acid and subjected to C18 solid-phase extraction (SPE) (Sep-Pak, Waters).

## Proteomics–off-line basic pH reversed-phase (BPRP) fractionation

Dried TMTpro-labeled sample was resuspended in 100 μl of 10 mM $NH_4HCO_3$, pH 8.0, and fractionated using BPRP HPLC. Briefly, samples were offline fractionated over 90 min and separated into 96 fractions by high pH reverse-phase HPLC (Agilent LC1260) through an Agilent ZORBAX 300Extend C18 column (3.5-μm particles, 4.6-mm ID and 250 mm in length) with mobile phase A containing 5% acetonitrile and 10 mM $NH_4HCO_3$ in LC-MS grade $H_2O$, and mobile phase B containing 90% acetonitrile and 10 mM $NH_4HCO_3$ in LC-MS grade $H_2O$ (both pH 8.0). The 96 resulting fractions were then pooled in a non-continuous manner into 24 fractions, and 12 fractions (non-adjacent) were used for subsequent MS analysis. For lysosomal extracts, the dried TMTpro-labeled sample was resuspended in 300 μL of 0.1% trifluoroacetic acid and then fractionated into six fractions, using the high pH reversed-phase peptide fractionation kit (Thermo Fisher Cat#84868). Fractions were vacuum centrifuged to near dryness. Each consolidated fraction was desalted via StageTip, dried again via vacuum centrifugation, and reconstituted in 5% acetonitrile, 1% formic acid for LC-MS/MS processing.

## Proteomics–liquid chromatography and tandem mass spectrometry

MS data were collected using an Orbitrap Fusion Lumos MS (Thermo Fisher) coupled to a Proxeon EASY-nLC1200 LC pump (Thermo Fisher). Peptides were separated on a 100-μm inner diameter microcapillary column packed in house with ~35 cm of Accucore150 resin (2.6 μm, 150 Å, Thermo Fisher) with a gradient consisting of 5–21% (0–125 min), 21–28% (125–140 min) (ACN, 0.1% FA) over a total 150 min run at ~500 nL/min. For analysis, we loaded 2–3 μg of each fraction onto the column. Each analysis used the Multi-Notch $MS^3$-based TMT method, to reduce ion interference compared to $MS^2$ quantification. The scan sequence began with an $MS^1$ spectrum (Orbitrap analysis; resolution 120,000 at 200 Th; mass range 400–1400 $m/z$; automatic gain control (AGC) target $5 \times 10^5$; maximum injection time 50 ms). Precursors for $MS^2$ analysis were selected using a Top10 method. $MS^2$ analysis consisted of collision-induced dissociation (quadrupole ion trap analysis; Turbo scan rate; AGC $2.0 \times 10^4$; isolation window 0.7 Th; normalized collision energy (NCE) 35; maximum injection time 35 ms). Monoisotopic peak assignment was used, and previously interrogated precursors were excluded using a dynamic window (120 s ± 10 ppm). After acquisition of each $MS^2$ spectrum, a synchronous-precursor-selection $MS^3$ scan was collected on the top 10 most intense ions in the $MS^2$ spectrum[34]. $MS^3$ precursors were fragmented by high-energy collision-induced dissociation and analyzed using the Orbitrap (NCE 55; AGC $3 \times 10^5$; maximum injection time 100 ms, resolution was 50,000 at 200 Th).

## Proteomics–data analysis

Raw mass spectra obtained were processed using Sequest. Mass spectra were converted to mzXML using a version of ReAdW.exe. Database searching included all entries from the Human Reference Proteome. Searches were performed with the following settings: (1) 20 ppm precursor ion tolerance for total protein level analysis, (2) Product ion tolerance was set at 0.9 Da, (3) TMTpro on lysine residues or N-termini at +304.207 Da, and (4) Carbamidomethylation of cysteine residues (+57.021 Da) as a static modification and oxidation of methionine residues (+15.995 Da) as a variable modification. Peptide-spectrum matches (PSMs) were adjusted to a 1% false discovery rate[50] PSM filtering was performed using a linear discriminant analysis. To quantify the TMTpro-based reporter ions in the datasets, the summed signal-to-noise (S:N) ratio for each TMTpro channel was obtained and found the closest matching centroid to the expected mass of the TMTpro reporter ion (integration tolerance of 0.003 Da). Proteins were quantified by summing reporter ion counts across all matching PSMs, as described[51] PSMs with poor quality, or isolation specificity <0.7, or with TMTpro reporter summed signal-to-noise ratio that were less than 100 or had no $MS^3$ spectra were excluded from quantification. Values for protein quantification were exported and processed using Perseus to calculate Log fold-changes and p-values. Volcano plots using these values were plotted in Excel.

## Thin-layer chromatography analysis

Analytic thin-layer chromatography (TLC) was performed on 10-cm high-performance thin-layer chromatography (HPTLC) plates (Sigma, Cat# 1056310001). The organic fractions of samples were dried down and analyzed by 2D TLC with chloroform:methanol:water-concentrated ammonia 70:30:3:2 (by vol) used as the first dimension and chloroform:methanol:water 65:35:5 (by vol) used as the second dimension as described[52]. Standard lipids (10 μg) dissolved in methanol or chloroform-methanol (1:1, v/v) were used as a reference.

## Statistical analysis

All statistical analysis was performed using GraphPad Prism 8. Information about significance test is provided in the respective figure

legends. All multiple comparisons were performed with the Dunn multiple comparisons correction.

### Reporting summary

Further information on research design is available in the Nature Research Reporting Summary linked to this article.

## Data availability

The mass spectrometry proteomics raw data generated in this study have been deposited in the ProteomeXchange Consortium via the PRIDE partner repository under accession code identifier PXD035889.

Lipidomics raw data has been deposited in Zenodo.com under identifier https://doi.org/10.5281/zenodo.6975484. The mass spectrometry data for all main and supplementary figures have been included in the source data file and in the Supplementary data files. Source data are provided with this paper.

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

## Acknowledgements

We thank members of the Farese and Walther laboratory, Gilbert Di Paolo and Todd Logan (Denali Therapeutics), Richard Ransohoff (Third Rock Ventures), and Konrad Sandhoff for helpful discussions. We thank Gary Howard for editorial services. This work was supported by the Bluefield Project to Cure FTD (S.B., J.W.H., T.C.W., and R.V.F.), NIH grants R01NS083524 (to J.W.H) and R01GM132129 (to J.A.P.), a sponsored research project from Google Ventures/Third Rock Ventures (J.W.H.), Aligning Science Across Parkinson's (ASAP-000282) initiative (to J.W.H.), the Canadian Institutes for Health Research (to S.S.), and the Howard Hughes Medical Institute (T.C.W.). The UCSF Neurodegenerative Disease Brain Bank receives funding support from NIH grants P30AG062422, P01AG019724, U01AG057195, and U19AG063911, as well as the Rain-water Charitable Foundation and the Bluefield Project to Cure FTD. The Michael J. Fox Foundation administers the grant ASAP-000282 on behalf of ASAP and itself. For the purpose of open access, the author has applied a CC-BY public copyright license to the Author Accepted Manuscript (AAM) version arising from this submission.

## Author contributions

T.C.W. and R.V.F. contributed equally as senior authors. S.B., R.V.F., and T.C.W. designed the study; S.B., S.S., P.C.M., A.D.N., G.A., and R.C.R. performed experiments; S.B. performed and analyzed lipidomics mass spectrometry experiments; Y.A.A., A.W.F., and S.S. contributed with sample preparations; A.L.N., S.S., L.T.G., and W.W.S. procured brain bank samples for biochemical analysis; S.B., M.S., and C.K. performed and analyzed lipid mass spectrometry experiments of human brain samples; S.S. performed gene editing and cloning; J.W.H. co-supervised cultured cell and proteomics experiments by S.S.; J.A.P provided mass spectrometry expertise for the proteomics experiments; S.B. and S.S. created figures and S.B., S.S., R.V.F., and T.C.W. wrote the manuscript with comments from all authors.

## Competing interests

J.W.H. is a founder and scientific advisory board member of Caraway Therapeutics Inc. and a founding scientific board member of Interline Therapeutics Inc. R.V.F. serves gratis on the board of the Bluefield Pro-ject to Cure FTD. T.C.W. is a founder and scientific advisory board chair of Antora Bio Inc. The remaining authors declare no competing interests.
