## [Peer Review File · Nature Communications]

REVIEWER COMMENTS

Reviewer #1 (Remarks to the Author):

This is a review of a very interesting manuscript entitled "Deficiency of the frontotemporal dementia gene GRN results in gangliosidosis" by Boland et al. Here, the authors address a longstanding and central question in the neurodegeneration field: Why does loss of the intralysosomal protein progranulin (PGRN) result in lysosomal dysfunction, likely the root cause of neurodegeneration in patients with GRN haploinsufficiency caused by familial mutations? Using a combination of studies in human GRN KO HeLa cells, mouse models, and human tissues, the authors carefully build a case for abnormalities in the processing of lysosomal lipids in settings of GRN deficiency. They observed that PGRN binds directly to a specific lipid, BMP, and that cells, mice, and humans with GRN deficiency have substantially lower levels of BMP, and subsequent gangliosidosis. Critically, they convincingly show that the abnormalities in lysosomal lipid composition is not directly caused by abnormal levels or activity of the majority of lysosomal enzymes.

Taken together, the discovery of dysregulated lipid homeostasis in GRN deficiency represents an important milestone in the field, and I believe give us a glimpse of the most proximal biology to GRN loss that has been described thus far. Their results suggest how dysregulation of lipid homeostasis within lysosomes may lead to eventual cellular dysfunction and disease, and interestingly suggest mechanistic links to GBA abnormalities that were suggested but not convincingly proven by prior studies. The findings by Boland et al complement and extend contemporaneous observations made by Logan et al, the latter of whom reductions observed reductions of BMP in mice and humans that could be rescued by a brain-penetrant PGRN chimeric protein. Importantly, the work by Boland et al also describes a robust human cell model that seems to recapitulate many of the abnormal lipid signatures of mouse models and human patients (and can be rescued by re-expression of PGRN). This cell model will be a key resource for the field for future mechanistic studies.

I have no major criticisms of the manuscript; experiments were thoughtfully designed and rigorously performed with proper controls and orthogonal validations. The paper itself is beautifully written and easy to read. My only minor question relates to the BMP-bead pulldown experiments: Did the authors also see binding of (cleaved) GRNs to the beads, and if so, were any particular GRNs more avidly bound to BMP? This would be interesting to test, given that it is likely that cleaved GRNs would interact with BMPs in acidic lysosomal environments.

Reviewer #2 (Remarks to the Author):

This manuscript by Boland et al reports a link between progranulin (PGRN), a lysosomal protein whose gene is mutated in frontotemporal dementia (FTD), and ganglioside metabolism. The authors use LC/MS techniques to demonstrate loss of PGRN in vitro (HeLa cells), in vivo (GRN knock-in mice) or in the brain of FTD patients perturbs ganglioside metabolism, resulting in accumulation of different species depending on the specimen used. They also confirmed data from the literature showing that PGRN loss causes a deficiency of the lysosomal phospholipid, BMP in KO cells, KO/KI tissues and FTD patient samples. The authors argue that BMP deficiency may be the mechanism underlying the ganglioside accumulation.

Overall, this is an important manuscript that demonstrates PGRN regulates ganglioside metabolism and that FTD brain is associated with ganglioside storage, similar to many lysosomal storage disorders. The study lends more support to the hypothesis a key function of PGRN may be to regulate lysosomal lipid catabolism. Mechanistically, the authors propose that the deficiency of BMP observed in PGRN loss of function models

may be responsible for ganglioside accumulation, although they do not provide any data demonstrating causality. Additionally, the BMP deficiency was reported before, so this should not be described as a novel finding in the paper, including in the abstract. In light of these issues, the paper needs to include more mechanistic data explaining the potential relationship between PGRN, BMP and ganglioside dysregulation to make it worthy of Nature Communications. This may be accomplished by addressing the following points.

Key points:

- 1. BMP deficiency appears to be an essential feature of PGRN LOF although the precise mechanism underlying this phenotype is unclear. Since BMP is believed to be in intraluminal vesicles (ILVs) within the endolysosomal compartment, could the mechanism be a reduced number or density of ILVs in GRN KO cells? Do these cells have fewer multivesicular bodies based on EM analyses?**
- 2. If BMP deficiency is responsible for ganglioside accumulation in GRN KO cells, can the addition of synthetic BMP liposomes rescue this phenotype? This is essential to establish causality.**
- 3. Does BMP regulate ganglioside metabolism via modulation of the lipases themselves or via accessory proteins like saposins or GM2 activators? Can the authors provide more mechanistic insights into this regulation?**
- 4. The authors claim that ganglioside accumulation is pathological in PGRN LOF cells. Can they actually show improved lysosomal function after correction of ganglioside storage in GRN KO cells with recombinant (or overexpressed) ganglioside catabolizing enzymes?**
- 5. The authors argue that FTD-GRN patients do not have known peripheral manifestations (despite the fact PGRN is ubiquitously expressed) and hypothesize that this is because gangliosides are enriched in the CNS. They also show ganglioside data from GRN KO/KI brains. Can the authors also assess ganglioside dysregulation in GRN KO/KI peripheral tissues, such as the liver? In other words, is the ganglioside accumulation specific for the CNS?**
- 6. Several labs have also described a critical role for BMP in facilitating cholesterol egress from lysosomes. The Storch lab showed it occurred at least in part through the interaction of BMP with NPC2. Since BMP levels are down in the PGRN LOF, does it cause any cholesterol phenotypes, including redistribution in lysosomes like in NPC cells? If so, might cholesterol accumulation cause secondary storage of gangliosides, as it does in NPC?**
- 7. Is lysosomal delivery required for exogenously expressed PGRN to rescue the ganglioside phenotype in GRN KO cells? For instance, does PGRN require sortilin or prosaposin binding for the rescue of KO-associated ganglioside phenotypes?**

Additional points:

- 8. The authors mention changes in the levels of PEs, cardiolipin, sphingomyelin, sterol esters and TAGs in the frontal lobes of FTD GRN patients – could the potential significance of these changes and their role in the pathologies of FTD-GRN be discussed?**
- 9. A recent paper (PMID 34450028) demonstrated that BMP deficiency in GRN KO cells causes accumulation of GBA substrate, glucosylsphingosine. Since the proposed mechanism presented in that published manuscript is highly similar to the one presented in the Boland et al around gangliosides, it should be discussed in this manuscript, perhaps even integrated into their model in the last figure.**

10. The analyses conducted in the brain of healthy controls and FTD patients are interesting but very much underpowered. For instance, it seems that they only use 3 healthy controls. Can the authors include more healthy control brains in their analysis or at the very least acknowledge that is a limitation of their study? Also, what is the specific brain region analyzed?

11. As mentioned above, the BMP findings obtained from PGRN LOF models should not be described as novel findings in this manuscript, but rather recognized as part of the general background from the literature highlighting a role for PGRN in lysosome function.

Reviewer #3 (Remarks to the Author):

The manuscript "Deficiency of the frontotemporal dementia gene GRN results in gangliosidosis" by Boland et al. describes an intriguing observation, i.e. lysosomal accumulation of gangliosides in brain of PGRN-deficient mice. This abnormality seems likely to be ascribed to the noted reduction of the lysosomal lipid BMP known to assist ganglioside degradation in lysosomes. This finding is highly noteworthy. It may shed new light on the pathophysiological mechanism underlying the outcome of PGRN abnormalities: homozygosity of mutant PGRN linked a form of neuronal ceroid lipofuscinosis and haploinsufficiency of GRN being associated with frontotemporal dementia (FTD).

The study is introduced in concise but clear manner. The experiments are motivated well and are conducted with state-of-the-art methods.

A weakness of the present manuscript is the lack of attention to glucosylceramide, the penultimate product in lysosomal catabolism of glycosphingolipids. This is even more so since the authors point out that GBA (lysosomal glucocerebrosidase catalyzing the degradation of glucosylceramide) is variably abnormal GRN^{-/-} cell lysates (an observation earlier made by others). Furthermore, the degradation of glucosylceramide is known to be assisted by BMP (seminal work by Sandhoff and colleagues, reference 20 manuscript). The noted deficiency of BMP during PGRN deficiency might therefore conceivably also cause abnormalities in glucosylceramide that on their turn could play a role in pathophysiology.

In fact, the authors do report on HexCer (hexosylceramide) levels which is the sum of galactosylceramide and glucosylceramide. It is nowadays very well feasible to separate glucosylceramide and galactosylceramide with a HILIC column and thus quantify separately the glucosylceramide and galactosylceramide (an abundant brain lipid). Inclusion of data on glucosylceramide would have been informative and have further strengthened the manuscript.

Reviewer #4 (Remarks to the Author):

The manuscript studied lysosomal lipid degradation influenced by progranulin deficiency in frontotemporal dementia. Gangliosides and other lipids were quantified in mouse and human brains with PGRN deficiency. The authors then thoroughly investigated various lysosomal functions and enzymes involved in GSL degradation in HeLa cells with GRN KO, Ctrl, and GRN KO-add back. The manuscript was well written with enough molecular details in searching for the reasons of impaired ganglioside degradation.

Overall, the manuscript provided a unique angle and very useful information about lysosomal function in lipid degradation involved in PGRN deficiency and FTD. This is very important to the field and fits well with the scope of the journal. But before it gets

accepted, the authors should address several major comments as listed below:

1. The author showed the overall changes of different types of lipids in Figure 2 and Extended Figure 1 (eg. PC, PE, PS, DAG, TAG...). How many lipid molecules were identified and used for quantification for each type of lipid? The quantification results for all the identified lipid molecules including the types of these lipids should be provided in supplemental tables.

2. In results section page 3, FTD patient brain tissues showed region-specific changes of lipids in frontal lobes vs. occipital lobe. Did the mouse model present such region-specific lipid changes? Was the lipidomics experiments of mouse brain conducted from the whole mouse brain or specific region of the mouse brain?

3. In Figure 2d and extended Figure 2B, the author showed that not all cells contained GM2 puncta. But extended Figure 2B also showed that some cells with GM2 puncta does not contain LAMP1 puncta, which is odd given that GM2 should mostly accumulate within the lysosomes. Can the authors provide individual panel of GM2 and LAMP1 staining beside the merged image for better clarity and also quantify the percentage of GM2 puncta that overlap with LAMP1?

4. For Figure 3F, the authors stated that LLOMe treatment significantly increased glectin 3 in PGRN KO vs. Ctrl cells. However, in Figure 3F, not all cells in the view showed such change. Can the author provide a quantification result of glectin3 and Ub punta, and glectin 3 & UB western blotting results to support such claim?

5. In page 5, it is not clear what is the specific role of BMP in degrading GSL and ganglioside in lysosomes. Can the author provide more explanation of BMP's function or specific molecular pathway it involves in gangliosidosis?

6. It is quite interesting to discover consistent reduction of BMP in HeLa cells, mouse and human brains with PGRN deficiency. But the conclusion that PGRN may protect BMP from degradation, and that low BMP levels result in gangliosidosis are a bit overreaching. The fact that PGRN interact with BMP does not prove that it protects BMP from degradation. The results and discussion section about relationships of BMP and PGRN also fall short in the manuscript. Can the author provide more evidence or at least discussions or future directions to further elucidate the specific roles of BMP in PGRN deficiency?

7. Protein identification and quantification results from HeLa cells and brain tissues should be provided in supplemental tables.

8. A data availability statement needs to be included and the lipidomics and proteomics raw data should be deposited to online repository like MassIVE or PRIDE.

REVIEWER COMMENTS:

Reviewer #1:

This is a review of a very interesting manuscript entitled “Deficiency of the frontotemporal dementia gene GRN results in gangliosidosis” by Boland et al. Here, the authors address a longstanding and central question in the neurodegeneration field: Why does loss of the intralysosomal protein progranulin (PGRN) result in lysosomal dysfunction, likely the root cause of neurodegeneration in patients with GRN haploinsufficiency caused by familial mutations? Using a combination of studies in human GRN KO HeLa cells, mouse models, and human tissues, the authors carefully build a case for abnormalities in the processing of lysosomal lipids in settings of GRN deficiency. They observed that PGRN binds directly to a specific lipid, BMP, and that cells, mice, and humans with GRN deficiency have substantially lower levels of BMP, and subsequent gangliosidosis. Critically, they convincingly show that the abnormalities in lysosomal lipid composition is not directly caused by abnormal levels or activity of the majority of lysosomal enzymes.

The discovery of dysregulated lipid homeostasis in GRN deficiency represents an important milestone in the field, and I believe give us a glimpse of the most proximal biology to GRN loss that has been described thus far. Their results suggest how dysregulation of lipid homeostasis within lysosomes may lead to eventual cellular dysfunction and disease, and interestingly suggest mechanistic links to GBA abnormalities that were suggested but not convincingly proven by prior studies. The findings by Boland et al complement and extend contemporaneous observations made by Logan et al, the latter of whom reductions observed reductions of BMP in mice and humans that could be rescued by a brain-penetrant PGRN chimeric protein. Importantly, the work by Boland et al also describes a robust human cell model that seems to recapitulate many of the abnormal lipid signatures of mouse models and human patients (and can be rescued by re-expression of PGRN). This cell model will be a key resource for the field for future mechanistic studies.

I have no major criticisms of the manuscript; experiments were thoughtfully designed and rigorously performed with proper controls and orthogonal validations. The paper itself is beautifully written and easy to read. My only minor question relates to the BMP-bead pulldown experiments: Did the authors also see binding of (cleaved) GRNs to the beads, and if so, were any particular GRNs more avidly bound to BMP? This would be interesting to test, given that it is likely that cleaved GRNs would interact with BMPs in acidic lysosomal environments.

We thank this reviewer for her/his helpful evaluation of our manuscript and the thoughtful suggestions. In response to the reviewer’s critique, our manuscript has undergone a major revision.

The reviewer suggests a very good experiment, and we agree that the interaction of GRNs with BMP warrants further investigation.

To address these questions, we first performed more rigorous biochemical binding assays of PGRN binding to lipids with liposome floatation assays (now shown in Suppl Fig. 4). We again found that a minor fraction of PGRN bound to BMP-containing but not PC-containing

liposomes. However, we also found that the majority of HIS-tagged PGRN bound to BMP-containing liposomes. Similarly, a control protein, HIS-tagged Cas9, efficiently bound to negatively charged liposomes. At pH 4.5, His (pKa 6.0) is protonated and positively charged, and BMP (pKa between 1.0 and 3.0) is negatively charged, suggesting that a HIS tag may mediate binding to BMP-containing liposomes. Our data support the hypothesis that the HIS-tag confers most of the binding that we observe. We note that the previous report of PGRN binding to BMP liposomes at acidic pH (Logan et al.) also utilized recombinant PGRN with a HIS-tag. We present these data in the revised manuscript and conclude that PGRN may bind directly to BMP, but that this requires further investigation.

With respect to granulins, we performed several experiments expressing different granulins and testing binding in the liposome flotation assays. Although in some experiments we found results consistent with some BMP-liposome binding, those results were not sufficiently convincing that we would feel comfortable with their publication. We therefore think this question will require more extensive studies beyond the scope of this report, as there are many variables including the epitope tags, sufficient linker regions, proper granulin folding and more.

Reviewer #2

This manuscript by Boland et al reports a link between progranulin (PGRN), a lysosomal protein whose gene is mutated in frontotemporal dementia (FTD), and ganglioside metabolism. The authors use LC/MS techniques to demonstrate loss of PGRN in vitro (HeLa cells), in vivo (GRN knock-in mice) or in the brain of FTD patients perturbs ganglioside metabolism, resulting in accumulation of different species depending on the specimen used. They also confirmed data from the literature showing that PGRN loss causes a deficiency of the lysosomal phospholipid, BMP in KO cells, KO/KI tissues and FTD patient samples. The authors argue that BMP deficiency may be the mechanism underlying the ganglioside accumulation.

Overall, this is an important manuscript that demonstrates PGRN regulates ganglioside metabolism and that FTD brain is associated with ganglioside storage, similar to many lysosomal storage disorders. The study lends more support to the hypothesis a key function of PGRN may be to regulate lysosomal lipid catabolism. Mechanistically, the authors propose that the deficiency of BMP observed in PGRN loss of function models may be responsible for ganglioside accumulation, although they do not provide any data demonstrating causality. Additionally, the BMP deficiency was reported before, so this should not be described as a novel finding in the paper, including in the abstract. In light of these issues, the paper needs to include more mechanistic data explaining the potential relationship between PGRN, BMP and ganglioside dysregulation to make it worthy of Nature Communications. This may be accomplished by addressing the following points.

We thank this reviewer for her/his critical and helpful evaluation of our manuscript. In response to the reviewer's critique, our manuscript has undergone a major revision.

Key points:

1. BMP deficiency appears to be an essential feature of PGRN LOF although the precise mechanism underlying this phenotype is unclear. Since BMP is believed to be in intraluminal vesicles (ILVs) within the endolysosomal compartment, could the mechanism be a reduced number or density of ILVs in GRN KO cells? Do these cells have fewer multivesicular bodies based on EM analyses?

We thank the reviewer for this interesting suggestion and, in response, have now performed electron microscopy of KO cells. These images did not reveal any marked differences among control, GRN KO and GRN KO cells with respect to ILV number/size.

As an alternative assay, we performed western blots and lysosomal proteomics of control cells, GRN KO cells, and GRN KO+GRN cells to examine a marker of ILVs, LAPT4A/B. From the isolation and proteomics analysis of lysosomes, we found that the abundance of several ILV-associated proteins (e.g., LAPT4A, LAPT4B, and SDCBP) are reduced in GRN KO cells, relative to control cells. We performed western blotting from cellular lysates to demonstrate that the decrease in LAPT4B levels in GRN KO cells can be restored through the genetic reintroduction of GRN. Lastly, we demonstrate that the decreased abundance of the ILV-associated marker protein LAPT4B in GRN KO HeLa cells is phenocopied in mouse brains deficient for GRN. These results are presented in the revised manuscript.

Based on these results, we now hypothesize that the number of ILVs may not be affected, but that their protein/lipid composition is altered, contributing to impaired lipid/ganglioside hydrolysis inside the lysosomes.

2. If BMP deficiency is responsible for ganglioside accumulation in GRN KO cells, can the addition of synthetic BMP liposomes rescue this phenotype? This is essential to establish causality.

We thank the reviewer for suggesting this important experiment. We employed two orthogonal approaches feeding different lipid species to control and GRN KO cells and measured GM2 gangliosides using IF and mass spectrometry.

Indeed, addition of BMP(18:1) liposomes rescued the GM2 levels to control cell levels in GRN KO cells by the IF assay (now shown in Suppl. Fig 4c and d). Similarly, addition of BMP(18:1) but not PC(18:1) normalized levels of gangliosides in GRN KO cells, as measured by mass spectrometry (now shown in Figure 4e).

3. Does BMP regulate ganglioside metabolism via modulation of the lipases themselves or via accessory proteins like saposins or GM2 activators? Can the authors provide more mechanistic insights into this regulation?

This is a good question. To address it, we performed additional immunoblots for prosaposin and GM2AP and found no differences in amounts of these proteins in cell lysates. Additionally, these proteins, as well as the relevant lipases, are shown unchanged in the main Figures 2a-d in proteomics of isolated lysosomes from WT and KO cells.

Therefore, the current data do not support that the amounts of these proteins are altered. Konrad Sandhoff's group proposed that BMP works by assisting in the binding of GSL degradative enzymes to ILVs in the lysosome and is essential for their function. We clarified these issues in the revised manuscript and model.

4. The authors claim that ganglioside accumulation is pathological in PGRN LOF cells. Can they actually show improved lysosomal function after correction of ganglioside storage in GRN KO cells with recombinant (or overexpressed) ganglioside catabolizing enzymes?

This is an interesting idea. However, we wish to point out that we detected correction of the gangliosidosis by restoring PGRN expression. We believe this is the most direct readout for restoring lysosomal activity, as other aspects of lysosomal function would be indirect and also appear to be less affected by lack of GRN.

5. The authors argue that FTD-GRN patients do not have known peripheral manifestations (despite the fact PGRN is ubiquitously expressed) and hypothesize that this is because gangliosides are enriched in the CNS. They also show ganglioside data from GRN KO/KI brains. Can the authors also assess ganglioside dysregulation in GRN KO/KI peripheral tissues, such as the liver? In other words, is the ganglioside accumulation specific for the CNS?

To address this, we measured gangliosides in kidney samples isolated from control, R493X PGRN HET and R493X PGRN KI mice. Indeed, similar to the findings in the mouse brain samples, deficiency of PGRN in kidney leads to elevated levels of gangliosides. However, the absolute levels of gangliosides in peripheral tissues are generally 2-10% of the levels in brain (PMID: 5822411), which may account for why the brain is particularly susceptible to PGRN deficiency. These results and additional discussion have been added to the text of the revised manuscript (Supplemental figure EDF1b).

6. Several labs have also described a critical role for BMP in facilitating cholesterol egress from lysosomes. The Storch lab showed it occurred at least in part through the interaction of BMP with NPC2. Since BMP levels are down in the PGRN LOF, does it cause any cholesterol phenotypes, including redistribution in lysosomes like in NPC cells? If so, might cholesterol accumulation cause secondary storage of gangliosides, as it does in NPC?

To address this question, we measured free cholesterol and cholesterol esters in genetically modified HeLa cell lines. We found no evidence of increased free cholesterol or reduced cholesterol esters in whole-cell lysates of PGRN KO cells, in contrast to the changes we found NPC1- or NPC2-deficient cell lines. These data are now reported in Supplemental Figure EDF2b. Additionally, we also could not detect any significant changes in free cholesterol in the frontal lobes of FTD GRN patients. Hence, we currently do not have evidence that cholesterol accumulation is the cause for the gangliosidosis.

7. Is lysosomal delivery required for exogenously expressed PGRN to rescue the ganglioside phenotype in GRN KO cells? For instance, does PGRN require sortilin or prosaposin binding for the rescue of KO-associated ganglioside phenotypes?

These are good questions. We do believe that exogenously expressed PGRN is trafficked to lysosomes and processed into granulins. The genetic reintroduction of PGRN into GRN KO cells rescues the levels of BMP and LAPT4B, an ILV-associated lipid and protein, respectively, that are found exclusively in lysosomes. Moreover, we quantified the change in abundance of PGRN peptides from proteomic analysis of purified lysosomes from control, GRN KO, and rescue cells, which confirmed the protein reaches the lysosome (not shown). Further, staining for GRN in control and addback lines shows ~equivalent levels of GRN that overlap with LAMP1 in lysosomes. Therefore, it is most likely that the rescue occurs due to lysosomal rescue.

How PGRN is trafficked to the lysosome has been investigated by several labs and appears to be via multiple pathways (sortilin, prosaposin, and other routes). Therefore, to dissect this further is a fairly major endeavor, and we believe beyond the major conclusions of the current manuscript.

Additional points:

8. The authors mention changes in the levels of PEs, cardiolipin, sphingomyelin, sterol esters and TAGs in the frontal lobes of FTD GRN patients – could the potential significance of these changes and their role in the pathologies of FTD-GRN be discussed?

We discuss this more extensively in the revised text.

9. A recent paper (PMID 34450028) demonstrated that BMP deficiency in GRN KO cells causes accumulation of GBA substrate, glucosylsphingosine. Since the proposed mechanism presented in that published manuscript is highly similar to the one presented in the Boland et al around gangliosides, it should be discussed in this manuscript, perhaps even integrated into their model in the last figure.

Thank you for the suggestion. We have added a discussion of this finding to the revised manuscript.

10. The analyses conducted in the brain of healthy controls and FTD patients are interesting but very much underpowered. For instance, it seems that they only use 3 healthy controls. Can the authors include more healthy control brains in their analysis or at the very least acknowledge that is a limitation of their study? Also, what is the specific brain region analyzed?

We appreciated the suggestion but are unable to add more healthy controls. These samples are much more limited in the UCSF Brain Bank.

More precise locations with the brain of the brain samples we assayed are now provided in the Methods section.

11. As mentioned above, the BMP findings obtained from PGRN LOF models should not be

described as novel findings in this manuscript, but rather recognized as part of the general background from the literature highlighting a role for PGRN in lysosome function.

Although we performed these studies in parallel to the recently published studies, we have clarified in the revised manuscript that the BMP findings are novel only for the highly relevant human brain samples.

Reviewer #3

The manuscript “Deficiency of the frontotemporal dementia gene GRN results in gangliosidosis” by Boland et al. describes an intriguing observation, i.e. lysosomal accumulation of gangliosides in brain of PGRN-deficient mice. This abnormality seems likely to be ascribed to the noted reduction of the lysosomal lipid BMP known to assist ganglioside degradation in lysosomes. This finding is highly noteworthy. It may shed new light on the pathophysiological mechanism underlying the outcome of PGRN abnormalities: homozygosity of mutant PGRN linked a form of neuronal ceroid lipofuscinosis and haploinsufficiency of GRN being associated with frontotemporal dementia (FTD).

The study is introduced in concise but clear manner. The experiments are motivated well and are conducted with state-of-the-art methods.

A weakness of the present manuscript is the lack of attention to glucosylceramide, the penultimate product in lysosomal catabolism of glycosphingolipids. This is even more so since the authors point out that GBA (lysosomal glucocerebrosidase catalyzing the degradation of glucosylceramide) is variably abnormal GRN^{-/-} cell lysates (an observation earlier made by others). Furthermore, the degradation of glucosylceramide is known to be assisted by BMP (seminal work by Sandhoff and colleagues, reference 20 manuscript). The noted deficiency of BMP during PGRN deficiency might therefore conceivably also cause abnormalities in glucosylceramide that on their turn could play a role in pathophysiology.

In fact, the authors do report on HexCer (hexosylceramide) levels which is the sum of galactosylceramide and glucosylceramide. It is nowadays very well feasible to separate glucosylceramide and galactosylceramide with a HILIC column and thus quantify separately the glucosylceramide and galactosylceramide (an abundant brain lipid). Inclusion of data on glucosylceramide would have been informative and have further strengthened the manuscript.

We thank this reviewer for her/his critical and helpful evaluation of our manuscript. In response to the reviewer’s critique, our manuscript has undergone a major revision.

Thank you for the suggestion on analyzing glucosylceramide and galactosylceramide levels. We measured these lipids in WT, HET and KI mouse brains and found no differences in these lipids, which are now reported in Supplemental figure 1a. We also added discussion of this finding in the revised manuscript.

Reviewer #4

The manuscript studied lysosomal lipid degradation influenced by progranulin deficiency in frontotemporal dementia. Gangliosides and other lipids were quantified in mouse and human brains with PGRN deficiency. The authors then thoroughly investigated various lysosomal functions and enzymes involved in GSL degradation in HeLa cells with GRN KO, Ctrl, and GRN KO-add back. The manuscript was well written with enough molecular details in searching for the reasons of impaired ganglioside degradation.

Overall, the manuscript provided a unique angle and very useful information about lysosomal function in lipid degradation involved in PGRN deficiency and FTD. This is very important to the field and fits well with the scope of the journal. But before it gets accepted, the authors should address several major comments as listed below:

1. The author showed the overall changes of different types of lipids in Figure 2 and Extended Figure 1 (e.g., PC, PE, PS, DAG, TAG...). How many lipid molecules were identified and used for quantification for each type of lipid? The quantification results for all the identified lipid molecules including the types of these lipids should be provided in supplemental tables.

We thank this reviewer for her/his critical and helpful evaluation of our manuscript. In response to the reviewer's critique, our manuscript has undergone a major revision.

We thank the reviewer for the suggestion and now provide data on all identified and quantitated lipid molecules in the extended data files.

2. In results section page 3, FTD patient brain tissues showed region-specific changes of lipids in frontal lobes vs. occipital lobe. Did the mouse model present such region-specific lipid changes? Was the lipidomics experiments of mouse brain conducted from the whole mouse brain or specific region of the mouse brain?

The mouse data were for whole cortex. We clarified this in the revised manuscript.

3. In Figure 2d and extended Figure 2B, the author showed that not all cells contained GM2 puncta. But extended Figure 2B also showed that some cells with GM2 puncta does not contain LAMP1 puncta, which is odd given that GM2 should mostly accumulate within the lysosomes. Can the authors provide individual panel of GM2 and LAMP1 staining beside the merged image for better clarity and also quantify the percentage of GM2 punta that overlap with LAMP1?

We thank the reviewer for this suggestion. We re-examined control/GRN KO/addback cells that were stained with antibodies directed against GM2 or LAMP1 after 4% PFA fixation/0.02% saponin permeabilization. As before, more and larger GM2 puncta in GRN KO cells partially colocalized with the lysosomal marker LAMP1. The specificity of the anti-GM2 antibody was further tested by using HEXA KO cells, which display an even greater accumulation of GM2 puncta. These data are included in the revised manuscript (Fig. 2d).

4. For Figure 3F, the authors stated that LLOMe treatment significantly increased lectin 3 in PGRN KO vs. Ctrl cells. However, in Figure 3F, not all cells in the view showed such change. Can the

author provide a quantification result of glectin3 and Ub punta, and glectin 3 & UB western blotting results to support such claim?

Based on this query, we repeated this experiment several times with different concentrations of the membrane-damaging agent LLOMe. However, we found the results to be variable for unclear reasons. We therefore removed them from the revised manuscript.

5. In page 5, it is not clear what is the specific role of BMP in degrading GSL and ganglioside in lysosomes. Can the author provide more explanation of BMP's function or specific molecular pathway it involves in gangliosidosis?

We added more discussion on how BMP may help to facilitate GSL degradation.

6. It is quite interesting to discover consistent reduction of BMP in HeLa cells, mouse and human brains with PGRN deficiency. But the conclusion that PGRN may project BMP from degradation, and that low BMP levels result in gangliosidosis are a bit overreaching. The fact that PGRN interact with BMP does not prove that it protects BMP from degradation. The results and discussion section about relationships of BMP and PGRN also fall short in the manuscript. Can the author provide more evidence or at least discussions or future directions to further elucidate the specific roles of BMP in PGRN deficiency?

With respect to PGRN binding to BMP, we first performed more rigorous biochemical binding assays of PGRN binding to lipids in liposome floatation assays (now shown in Suppl Fig. 4). We again found that a minor fraction of PGRNs bound to BMP-containing but not PC-containing liposomes. However, we also found that the majority of HIS-tagged progranulin bound to BMP-containing liposomes. Similarly, a control protein, HIS-tagged Cas9, efficiently bound to negatively charged liposomes. At pH 4.5, His (pKa 6.0) is protonated and positively charged, and BMP (pKa between 1.0 and 3.0) is negatively charged, suggesting that a HIS tag may mediate binding to BMP-containing liposomes. Our data support the hypothesis that the HIS tag confers most of the binding that we observe. We note that the previous report of PGRN binding to BMP liposomes at acidic pH (Logan et. al) also utilized recombinant PGRN with a His tag. We present these data in the revised manuscript and conclude that PGRN may bind directly to BMP, but that this requires further investigation.

With respect to the causality relationship of gangliosidosis and BMP levels, we performed additional experiments to test if BMP could rescue the phenotype. We employed two orthogonal approaches feeding different lipid species to control and GRN KO cells and measured:

- a) GM2 gangliosides by IF, and
- b) ganglioside species by mass spectrometry.

Addition of BMP(18:1) but not PC(18:1) liposomes rescued the GM2 levels back to control cell levels in GRN KO cells in the IF assay (now shown in Suppl. Fig 4c and d). Similarly, addition of BMP(18:1) but not PC(18:1) normalized levels of gangliosides in GRN KO cells, as measured by mass spectrometry (now shown in Figure 4e).

7. Protein identification and quantification results from HeLa cells and brain tissues should be provided in supplemental tables.

This is now provided.

8. A data availability statement needs to be included and the lipidomics and proteomics raw data should be deposited to online repository like MassIVE or PRIDE.

This is now provided.

REVIEWERS' COMMENTS

Reviewer #1 (Remarks to the Author):

The authors did a superb job in responding to my question related to GRN interactions with BMP, importantly introducing additional controls that showed that the majority of the interactions between PGRN and BMP were driven by the his tag. Major revisions were included to address concerns from the other reviewers, and I now believe that the manuscript is substantially improved and suitable for publication.

Reviewer #2 (Remarks to the Author):

The authors have done a great job addressing all the questions and concerns. There are however specific text changes that need to be made for accuracy and/or as cautionary notes.

1. The authors mentioned that they did not have access to more than 3 healthy controls in their lipid assessments in FTD brains. This is a VERY small sample size for human brain studies, so it is highly recommended that this caveat be noted in the main text.
2. The authors mentioned that the previous point I raised (#9) was addressed, but I did not see it anywhere. Generally it is a good idea to mention specifically where revisions are made in the text, so it's easier for referees to track them. As requested in the first round of reviews, the authors should mention that BMP deficiency in the Grn KO was found to mediate, at least in part, a decrease in GCase activity and increase in GCase substrate, glucosylsphingosine (GlcSph). While this does not relate directly to the ganglioside findings described in the manuscript under consideration, GlcSph, like gangliosides, is a glycosphingolipid and conceptually, the BMP/GlcSph relationship is very similar to the scenario the authors have in mind for the BMP/ganglioside relationship.
3. Regarding the possibility that the 6XHIS tag may bind to BMP at acidic pH, this is an important potential caveat to mention from the previous literature, but the authors' experiments are also confounded by other factors. It appears as though the authors may have used two completely different sources of recombinant progranulin to assess the impact of 6-HIS binding (one commercial source vs. one made in-house). Without proper quality control of the actual proteins, it is hard to know whether the folding of these two preparations is identical or whether other factors, besides the presence or absence of 6-HIS, contribute to the differential binding to BMP. Ideally, this experiment should be performed with the same source of recombinant progranulin, one with the tag, and another one with the 6-HIS tag cleaved off. In essence, the authors should mention specifically this caveat in the main text (i.e., that the untagged and 6-HIS tagged recombinant proteins are from different sources). Finally, 6xHIS CAS9 is potentially a bad negative control to use, because DNA binding proteins typically bind to DNA via basic residues, so it is extremely likely that CAS9 actually binds to BMP at acidic pH. If the authors want to keep this control, they should also mention the possibility that CAS9 may bind to BMP directly, independently of the tag. Alternatively, they could use another control 6-HIS tagged protein that does not have any basic residues.

Reviewer #3 (Remarks to the Author):

The authors satisfactorily addressed the request for supplying additional information on glucosylceramide levels (rather than hexosylceramide concentrations). These data are now included in the manuscript and the results are integrated in the text. This revision covers my question. The revised manuscript has considerable relevance and should be considered for publication.

Reviewer #4 (Remarks to the Author):

The authors have addressed all my comments. Just a minor yet important note: Please make sure all lipidomics and proteomics raw data are deposited online and available to the public with a link/access ID provided in the manuscript.

POINT-BY-POINT RESPONSES TO REVIEWERS' COMMENTS

Reviewer #1 (Remarks to the Author):

The authors did a superb job in responding to my question related to GRN interactions with BMP, importantly introducing additional controls that showed that the majority of the interactions between PGRN and BMP were driven by the his tag. Major revisions were included to address concerns from the other reviewers, and I now believe that the manuscript is substantially improved and suitable for publication.

We thank the reviewer for the time and effort spent to improve our paper.

Reviewer #2 (Remarks to the Author):

The authors have done a great job addressing all the questions and concerns. There are however specific text changes that need to be made for accuracy and/or as cautionary notes.

We thank the reviewer for the time and effort spent to improve our paper.

1. The authors mentioned that they did not have access to more than 3 healthy controls in their lipid assessments in FTD brains. This is a VERY small sample size for human brain studies, so it is highly recommended that this caveat be noted in the main text.

This has been noted in the revised version.

2. The authors mentioned that the previous point I raised (#9) was addressed, but I did not see it anywhere. Generally it is a good idea to mention specifically where revisions are made in the text, so it's easier for referees to track them. As requested in the first round of reviews, the authors should mention that BMP deficiency in the Grn KO was found to mediate, at least in part, a decrease in GCcase activity and increase in GCcase substrate, glucosylsphingosine (GlcSph). While this does not relate directly to the ganglioside findings described in the manuscript under consideration, GlcSph, like gangliosides, is a glycosphingolipid and conceptually, the BMP/GlcSph relationship is very similar to the scenario the authors have in mind for the BMP/ganglioside relationship.

This was mentioned previously in the discussion. We have now expanded on this point.

3. Regarding the possibility that the 6XHIS tag may bind to BMP at acidic pH, this is an important potential caveat to mention from the previous literature, but the authors' experiments are also confounded by other factors. It appears as though the authors may have used two completely different sources of recombinant progranulin to assess the impact of 6-HIS binding (one commercial source vs. one made in-house). Without proper quality control of the actual proteins, it is hard to know whether the folding of these two preparations is identical or whether other factors, besides the presence or absence of 6-HIS, contribute to the differential binding to BMP. Ideally, this experiment should be performed with the same source of recombinant progranulin, one with the tag, and another one with the 6-HIS tag cleaved off. In essence, the authors should mention specifically this caveat in the main text (i.e., that the untagged and 6-HIS tagged recombinant proteins are from different sources).

Finally, 6xHIS CAS9 is potentially a bad negative control to use, because DNA binding proteins typically bind to DNA via basic residues, so it is extremely likely that CAS9 actually binds to BMP at acidic pH. If the authors want to keep this control, they should also mention the possibility that CAS9 may bind to BMP directly, independently of the tag. Alternatively, they could use another control 6-HIS tagged protein that does not have any basic residues.

We thank the reviewer for these comments on the PGRN-BMP flotation binding assay. After consideration, we agree with the reviewer that the experiment was not optimally controlled and, since it does not affect the major conclusions of our paper, have therefore elected to remove it from the manuscript. We have discussed this with the editor, who agrees with this plan. We agreed to keep the bead-binding assay in the paper, since it is a better controlled experiment.

Reviewer #3 (Remarks to the Author):

The authors satisfactorily addressed the request for supplying additional information on glucosylceramide levels (rather than hexosylceramide concentrations). These data are now included in the manuscript and the results are integrated in the text.

This revision covers my question. The revised manuscript has considerable relevance and should be considered for publication.

We thank the reviewer for the time and effort spent to improve our paper.

Reviewer #4 (Remarks to the Author):

The authors have addressed all my comments. Just a minor yet important note: Please make sure all lipidomics and proteomics raw data are deposited online and available to the public with a link/access ID provided in the manuscript.

We thank the reviewer for the time and effort spent to improve our paper. We will make sure of this.

Peer Review File

Deficiency of GRN, a frontotemporal dementia gene, results in
gangliosidosis